# Characterization of *BRCA1*-deficient premalignant tissues and cancers identifies *Plekha5* as a tumor metastasis suppressor

Jianlin Liu[1,2], Ragini Adhav[1,2], Kai Miao[1,2], Sek Man Su[1,2], Lihua Mo[1,2], Un In Chan[1,2], Xin Zhang[1,2], Jun Xu[1,2], Jianjie Li[1,2], Xiaodong Shu[1,2], Jianming Zeng [1,2], Xu Zhang[1,2], Xueying Lyu[1,2], Lakhansing Pardeshi[1,3], Kaeling Tan[1,3], Heng Sun[1,2], Koon Ho Wong [1,3], Chuxia Deng [1,2✉] & Xiaoling Xu [1,2✉]

Single-cell whole-exome sequencing (scWES) is a powerful approach for deciphering intra-tumor heterogeneity and identifying cancer drivers. So far, however, simultaneous analysis of single nucleotide variants (SNVs) and copy number variations (CNVs) of a single cell has been challenging. By analyzing SNVs and CNVs simultaneously in bulk and single cells of premalignant tissues and tumors from mouse and human *BRCA1*-associated breast cancers, we discover an evolution process through which the tumors initiate from cells with SNVs affecting driver genes in the premalignant stage and malignantly progress later via CNVs acquired in chromosome regions with cancer driver genes. These events occur randomly and hit many putative cancer drivers besides *p53* to generate unique genetic and pathological features for each tumor. Upon this, we finally identify a tumor metastasis suppressor *Plekha5*, whose deficiency promotes cancer metastasis to the liver and/or lung.

[1] Cancer Centre, Faculty of Health Sciences, University of Macau, Macau, SAR, China. [2] Centre for Precision Medicine Research and Training, Faculty of Health Sciences, University of Macau, Macau, SAR, China. [3] Genomics, Bioinformatics & Single Cell Analysis Core, Faculty of Health Sciences, University of Macau, Macau, SAR, China. ✉email: cxdeng@um.edu.mo; xiaolingx@um.edu.mo

Breast cancer-associated gene 1 (*BRCA1*) is a tumor suppressor gene encoding a large protein that is involved in many essential biological processes, including DNA damage repair[1], cell cycle checkpoints[1], chromatin remodeling[2], transcriptional regulation[3], and protein ubiquitination[4]. The deficiency of *BRCA1* induces severe genome instability, eventually leading to tumorigenesis[1,5–7]. *BRCA1* mutations account for almost one-quarter of cases of hereditary familial breast cancer in humans, and the estimated lifetime risk of developing breast cancer for women with *BRCA1* germline mutations is 40–80%[8]. The majority of *BRCA1*-deficient breast cancer cases are classified as triple-negative breast cancer based on the absence of expression of estrogen receptor (ER), progesterone receptor (PR), and HER2[9], and targeted therapy for this type of cancer is difficult. The poly (ADP-ribose) polymerase inhibitors (PARPi) have been approved by the U.S. Food and Drug Administration for the treatment of *BRCA1*-mutated tumors[10,11]; however, only a portion of patients respond to this treatment[10–12] and drug resistance can still reduce the effectiveness of this therapy[13]. There is a pressing need to better understand this type of cancer at the genomic level from the early premalignant stage through to tumor progression and metastasis stage in order to develop effective strategies for the prevention and treatment of *BRCA1*-associated breast cancer.

Driver events for each individual person, even different tumors in the same patient might be different and need to be elucidated. At present, the biology governing the initiation of *BRCA1*-associated tumors is not fully understood. Considering the essential role of *BRCA1* in maintaining genome integrity, it is believed that loss of *BRCA1* causes random mutations in the genome[7], which randomly activate oncogenes or inactivate tumor suppressor genes (TSGs) and can be accumulated in the specific contexts of individual ecosystems under Darwinian natural selection, promoting tumor initiation, formation, and metastasis. We have previously established a mouse model carrying mammary-specific disruption of the full length of *Brca1* (*Brca1co/co;MMTV-Cre* or *Brca1co/co;Wap-Cre*) (*Brca1*MKO), which develops mammary tumors through multiple progressive steps involving spontaneous mutations of *p53* and/or some other unidentified genes[5,14]. Using a genetic approach in which mutant mice carrying targeted disruptions of *p53*, *Atm*, *Chk2*, or *53BP1* are bred with *Brca1*-deficient mice, we found that these genes also play critical roles in various aspects of tumorigenesis[15–19], yet many other putative tumor suppressors and oncogenes have not been identified.

Bulk high-throughput DNA sequencing has been widely used for driver mutation identification in many different types of cancers[20,21]. However, microscopic tumors, tumor purity, and intratumor heterogeneity may cast limits on the utilization of bulk DNA sequencing. Recently, single-cell DNA sequencing has been used for revealing clonal evolution, intratumor heterogeneity, and driver mutations in cancer[22–25]. Single-cell whole-genome sequencing has been used for copy number variation (CNV) analysis in several studies[22,23]; however, confident single nucleotide variant (SNV) calling is limited by the expensive cost of high coverage of the whole genome in single-cell whole-genome sequence studies. Whole-exome sequencing (WES) was previously used to obtain high coverage for confident SNV calling in single-cell analysis[24,25]; however, single-cell CNV analysis using whole-exome data is limited and challenged by the use of conventional bulk CNV calling algorithms. The combination of SNV and CNV data of the same single cells from different tumor developmental stages will provide an alternative strategy and a different view for characterizing cancer.

To understand the genetic features and evolution of *BRCA1*-deficient breast cancers and identify their drivers, we perform both bulk tissue WES and single-cell WES (scWES) for *BRCA1*-deficient mammary glands and tumors, followed by validation with Sanger sequencing or droplet digital PCR (ddPCR) assays, as well as functional validation with clustered regularly interspaced short palindromic repeats (CRISPR) and CRISPR-associated protein 9 (Cas9)-mediated knockouts in vitro and in vivo. We demonstrate that tumors initiate from cells with random SNVs affecting driver genes in the premalignant stage and malignantly progress later via CNVs acquired in chromosome regions with many cancer driver genes, including *Plekha5*, which acts as a tumor metastasis suppressor.

## Results

**Distinct features of tumors associate with different drivers.** We have previously shown that mutation of *p53* accelerates mammary tumor formation in mice carrying mammary-specific disruption of *Brca1* (Supplementary Fig. 1a)[14]. However, whether other driver mutations also contribute to the breast cancer initiation and progression associated with *BRCA1* deficiency is not well understood. To investigate mutation patterns in individuals with *BRCA1* deficiency, we conducted WES of 23 tumors from *Brca1co/co;MMTV-Cre*, *Brca1co/co;MMTV-Cre;mG/mT* or *Brca1co/co;p53+/co;Wap-Cre* mice (Supplementary Data 1). WES with a mean depth of 236X coverage in the mouse exome region (Supplementary Data 1) identified a total of 597 somatic SNVs in all tumors, including synonymous, missense, splice site, and frameshift mutations, etc. (Fig. 1a; Supplementary Fig. 1b). C>A/G>T was the major mutation type in these tumors derived from *Brca1* mammary-specific knockout (*Brca1*MKO) mice with *p53* mutation (Supplementary Fig. 1c); however, several tumors derived from *Brca1* mammary-specific knockout (*Brca1*MKO) mice with wild-type *p53* showed predominance in C>T/G>A (Supplementary Fig. 1c). *BRCA1*-related signature 3, breast cancer-associated signatures 8 and 18, and pan-cancer signatures 5 were observed in these tumors (Fig. 1b)[26]. The number of somatic SNVs varied from 6 to 59 for each tumor, including 3 to 18 nonsynonymous SNVs of human homologous genes (nSNVs) (Supplementary Fig. 1d; Supplementary Data 2). In addition, we analyzed 31 human breast cancers with *BRCA1* germline mutations from the Wellcome Trust Sanger Institute (WSI) dataset[21]. Analysis of this dataset revealed a high number of somatic nonsynonymous SNVs, ranging from 19 to 171 (Supplementary Fig. 1e). Oncoplots with somatic SNVs showed a highly unique mutation pattern in each tumor with very different SNVs or mutated genes in both mice and human patients bearing *BRCA1* mutations (Fig. 1c, d). Analyses of CNV patterns in both mice and human patients were also carried out, and similarly, each individual exhibited different amplification or deletion patterns, although some common amplification regions existed (Fig. 1e; Supplementary Fig. 1f; Supplementary Data 3). This variability of CNV patterns among tumors was consistent with that observed in the *Brca1*-deficient mouse tumors from a previous study[27].

To identify drivers in each tumor, we focused on SNVs and classified mutated genes into three groups, including well-known TSGs or oncogenes (TSGs/oncogenes)[28–30] (https://bioinfo.uth.edu/TSGene, http://ongene.bioinfo-minzhao.org/, https://cancer.sanger.ac.uk/cosmic), predicted TSGs or oncogenes (PRE-TSGs/oncogenes)[31], and others (Fig. 2a, b). Our results showed that each tumor carried mutations in at least one well-known TSG/oncogene or predicted TSG/oncogene, such as *p53*, *Kras*, and *Nras*, and many tumors carried more than one (Fig. 2a; Table 1). No obvious differences in the number of TSGs/oncogenes or predicted TSGs/oncogenes were observed between tumors with or without *p53* mutations (Fig. 2a, b; Table 1). Notably, the predicted TSG *Plekha5* mutation (c.872G>A, p.Ser291Asn) was found in 153 liver metastatic tumor (LMT) but not in 153 primary tumor

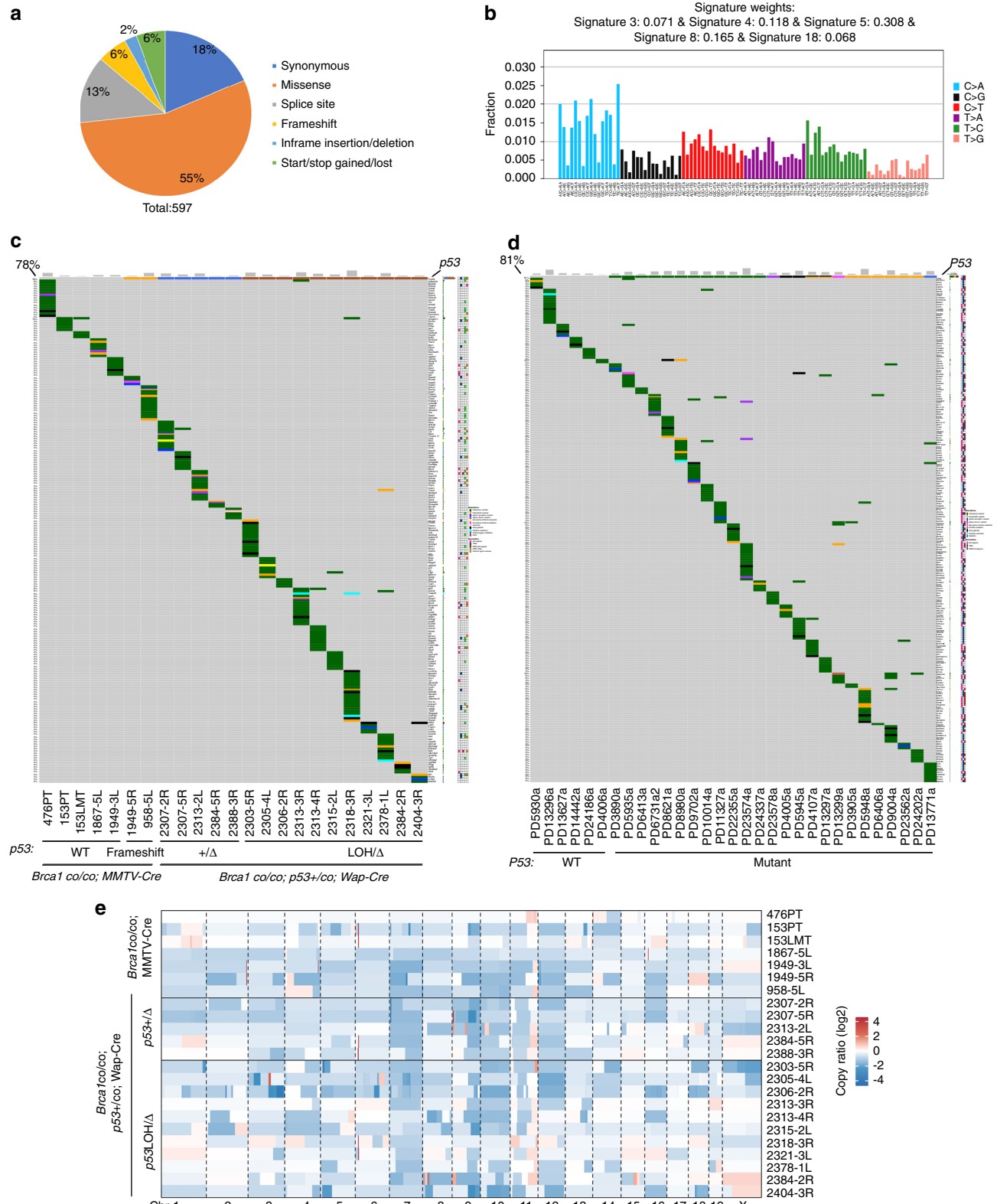

**Fig. 1 Unique mutation feature of _Brca1_-deficient breast tumors revealed by bulk WES. a** Mutation types in the coding exon region of _Brca1_-deficient breast tumors ($n = 23$ tumors). **b** Mutational signature in _Brca1_-deficient breast tumors ($n = 23$ tumors). Mutational signature was generated using all SNVs detected in samples. **c** Oncoplot summarizing all altered genes with nonsynonymous SNVs in _Brca1_-deficient breast tumors of _Brca1_^MKO mice ($n = 23$ tumors). PT-primary tumor, LMT-liver metastatic tumor, 5L-tumor in 5th left mammary gland, 5R-tumor in 5th right mammary gland, WT-wild type, $+/\Delta$-deletion of one allele of exon 5 and 6 of _p53_ mediated by Wap-Cre-LoxP, LOH/$\Delta$-deletion of two alleles of _p53_, one allele deletion is mediated by Wap-Cre-LoxP, another one is naturally lost. **d** Oncoplot summarizing all altered TSGs or oncogenes with nonsynonymous SNVs in breast tumors of patients with _BRCA1_ germline mutation from the WSI dataset ($n = 31$ tumors). **e** CNV profiles of _Brca1_-deficient breast tumors of _Brca1_^MKO mice ($n = 23$ tumors).

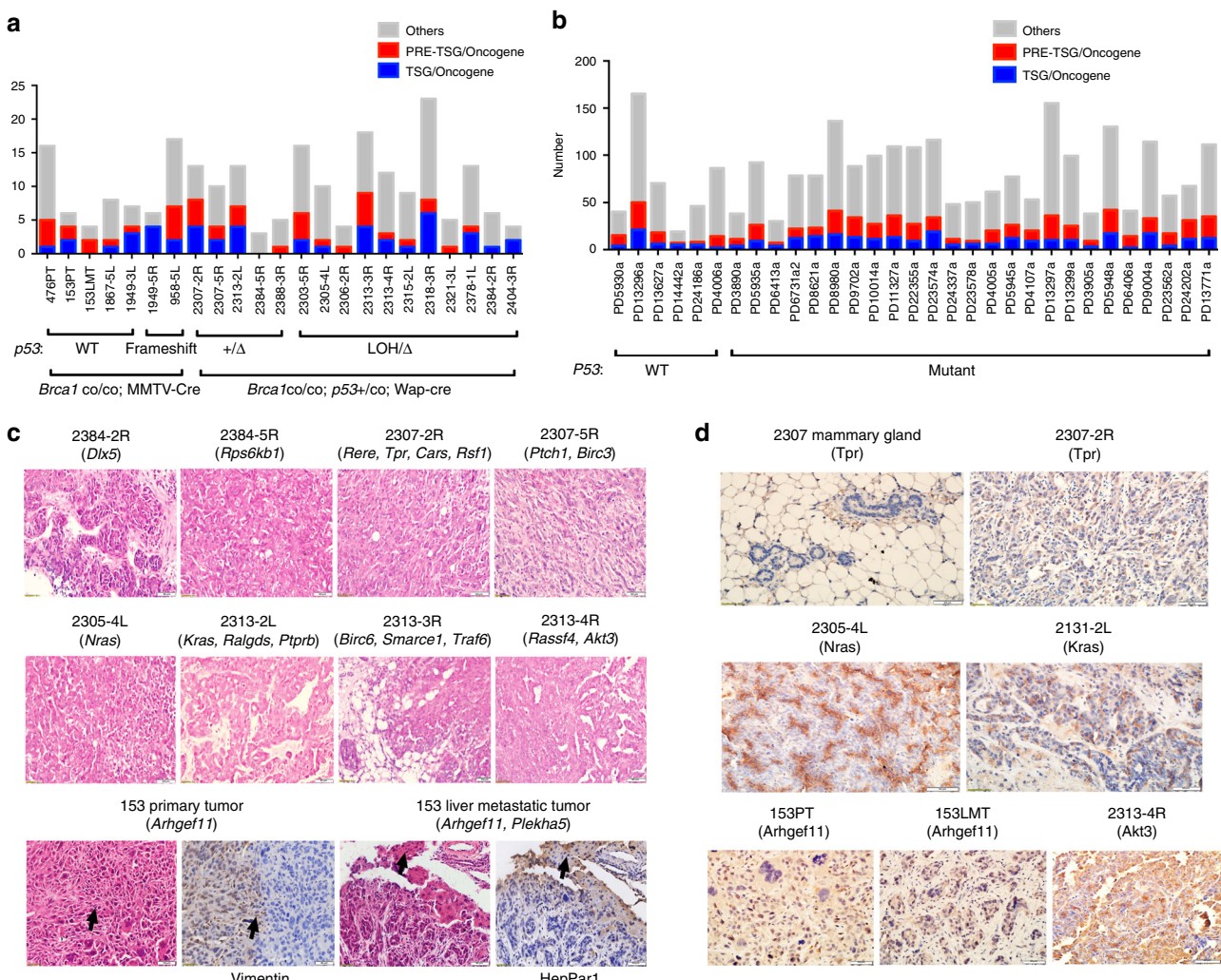

**Fig. 2 Identification of drivers showing high inter-tumor heterogeneity in *Brca1*-deficient tumors. a** Bar plot showing the number of TSGs/oncogenes and predicted TSGs/oncogenes mutated in each *Brca1*-deficient breast tumor of *Brca1*^MKO mice. **b** Bar plot showing the number of TSGs/oncogenes and predicted TSGs/oncogenes mutated in each breast tumor of patients with *BRCA1* germline mutation from the WSI dataset. **c** Histology and IHC staining showing the morphology in tumors carrying different potential driver mutations (indicated in the brackets). One representative sections were shown for each tumor. Scale bars, 50 μm. The bottom panel shows H&E and IHC staining in adjacent sections of 153PT and 153LMT. The first arrow of the bottom panel indicates aggressive mesenchymal tumor cells; the second arrow indicates aggressive mesenchymal tumor cells positive for EMT marker Vimentin; the third arrow indicates hepatocyte; the fourth arrow indicates hepatocyte positive for hepatocyte-specific marker HepPar1. **d** IHC staining of tumors using their corresponding potential driver oncogenes called from the WES data. All antibodies used are shown in the brackets. One representative section was shown for each tumor. Scale bars, 50 μm.

(PT) (Fig. 1c; Table 1). Potential driver mutations from 18 of 23 tumors were validated by Sanger sequencing (Supplementary Fig. 2a; Supplementary Data 4), and the remaining five tumors could not be validated due to the low variant allele fraction (VAF) of driver mutations (Table 1). The diverse VAFs of different driver mutations might be caused by tumor purity and intratumor heterogeneity, which may perturb the identification of real driver events.

Cre-LoxP-mediated deletion of exon 5–6 of *p53* (*p53* +/△) was found in all *Brca1^co/co^;p53^+/co^;Wap-Cre* mice, and loss of the remaining wild-type allele of *p53* (*p53* LOH/△) occurred in 11 out of 16 tumors (Table 1; Supplementary Fig. 2b), whereas frameshift mutations of *p53* were found in two out of seven tumors from five *Brca1^co/co^;MMTV-Cre* mice (Fig. 1c; Table 1). A total of 57 driver genes were found in 23 mouse tumor samples, and each tumor contained different cancer driver genes (Fig. 1c; Table 1).

Twenty-five of thirty-one (81%) breast cancer patients carrying *BRCA1* germline mutations had a *P53* mutation and a total of 231 potential driver genes were found in these breast cancer patients (Fig. 1d). These data suggest that while the loss of *P53* occurs in a large proportion of *BRCA1*-deficient breast cancer, additional tumor driver mutations are still necessary for tumorigenesis in a significant portion of the tumors in both mice and humans.

To further tie our sequencing data to the pathology of each individual tumor, we identified the driver genes for each corresponding tumor and found that different tumors from different mice presented distinct histopathology (Fig. 2c; Supplementary Fig. 2c). Next, we analyzed different tumors derived from the same mouse, including 2384-2R and 2384-5R; 2307-2R and 2307-5R; 2313-2L, 2313-3R, and 2313-4R, and 153PT and 153LMT, which showed distinct histopathology (Fig. 2c). These data are consistent with our sequencing data showing that each tumor carries different sets of driver genes.

**Table 1 Potential driver genes in each tumor.**

| Group | p53 status | Sample | Driver genes | TSG/oncogene | VAF | Days[e] |
|---|---|---|---|---|---|---|
| Brca1co/co; MMTV-Cre | Wild type | 476PT | Adam20 | PRE-TSG | 0.33 | 402 |
| | | | Prdm2 | TSG | 0.26 | |
| | | | Ttn | PRE-TSG | 0.23 | |
| | | | Ttc28 | PRE-TSG | 0.21 | |
| | | | Kansl2 | PRE-TSG | 0.21 | |
| | | 153PT | Arhgef11 | PRE-TSG | 0.37 | 440 |
| | | | Pbrm1 | TSG | 0.33 | |
| | | | Ipo7 | PRE-TSG | 0.24 | |
| | | | Psip1 | Oncogene | 0.21 | |
| | | 153LMT | Plekha5 | PRE-TSG | 0.52 | |
| | | | Arhgef11 | PRE-TSG | 0.28 | |
| | | 1867-5L | Esrp1 | TSG | 0.12 | 673 |
| | | 1949-3L | Aff3 | Oncogene | 0.37 | 651 |
| | | | Kdr | Oncogene | 0.13 | |
| | | | Cyb5a | TSG | 0.07 | |
| | Frameshift | 1949-5R | Trp53 | TSG | 0.87 | 651 |
| | | | Rasal1 | TSG | 0.46 | |
| | | | Hspd1 | TSG | 0.06 | |
| | | 958-5L | Trp53 | TSG | 0.82 | 929 |
| | | | Mms22l | Oncogene | 0.14 | |
| Brca1co/co; p53+/co; Wap-Cre | '+/△[a] | 2307-2R | Rere | Oncogene | 0.33 | 342 |
| | | | Tpr | Oncogene | 0.12 | |
| | | | Cars | TSG | 0.09 | |
| | | | Rsf1 | Oncogene | 0.06 | |
| | | 2307-5R[c] | Ptch1 | TSG | 0.16 | 342 |
| | | | Birc3 | Oncogene | 0.01 | |
| | | 2313-2L | Kras | Oncogene | 0.41 | 342 |
| | | | Ralgds | Oncogene | 0.31 | |
| | | | Ptprb | TSG | 0.12 | |
| | | | Ranbp9 | TSG | 0.05 | |
| | | 2384-5R | Rps6kb1 | [d] | 0.52 | 307 |
| | | 2388-3R[c] | Stx3 | PRE-TSG | 0.04 | 330 |
| | 'LOH/△[b] | 2303-5R | Abcg2 | TSG | 0.40 | 371 |
| | | | Atm | TSG | 0.03 | |
| | | 2305-4L | Nras | Oncogene | 0.80 | 342 |
| | | 2306-2R[c] | Top1 | PRE-TSG | 0.12 | 342 |
| | | 2313-3R[c] | Birc6 | Oncogene | 0.29 | 342 |
| | | | Smarce1 | TSG | 0.18 | |
| | | | Traf6 | Oncogene | 0.17 | |
| | | | Sf3b1 | Oncogene | 0.04 | |
| | | 2313-4R | Rassf4 | TSG | 0.15 | 342 |
| | | | Akt3 | Oncogene | 0.03 | |
| | | 2318-3R | Map4k1 | TSG | 0.37 | 365 |
| | | | Slit2 | TSG | 0.33 | |
| | | | Yy1 | Oncogene | 0.31 | |
| | | | Slx4 | TSG | 0.21 | |
| | | | Prkcb | TSG | 0.05 | |
| | | | Vegfa | TSG/oncogene | 0.04 | |
| | | 2321-3L | Lad1 | [d] | 0.39 | 342 |
| | | | Adh5 | PRE-TSG | 0.07 | |
| | | 2378-1L | Csmd1 | TSG | 0.41 | 330 |
| | | | Apc | TSG | 0.39 | |
| | | | Egfr | Oncogene | 0.14 | |
| | | 2384-2R[c] | Dlx5 | Oncogene | 0.07 | 307 |
| | | 2404-3R | Nf1 | TSG | 0.86 | 326 |
| | | | Shprh | TSG | 0.18 | |

[a]Deletion of one copy of exon 5 and 6 of p53 mediated by Wap-Cre-LoxP.
[b]Deletion of two copies of exon 5 and 6 of p53. One copy deletion is mediated by Wap-Cre-LoxP, another one is naturally lost.
[c]The driver mutations in this tumor cannot be validated by Sanger sequencing.
[d]The gene is neither a TSG/oncogene nor a PRE-TSG/Oncogene but with high VAF.
[e]Days to observe tumors.

To further validate drivers identified from each individual tumor, we performed immunohistochemistry (IHC) using the specific markers for tumors with corresponding driver genes. As expected, different tumors were positive for the specific driver genes; for example, 2305-4L has a *Nras* mutation (c.34G>C, p. Gly12Arg) and was positive for Nras, 2313-2L had a *Kras* mutation (c.35G>T, p.Gly12Val) and was positive for Kras, 2313-4R had an *Akt3* mutation (c.463G>T, p.Gly155Cys) and was positive for Akt3, 2307-2R had a *Tpr* (Translocated Promoter Region) mutation (c.3920G>T, p.Arg1307Leu) and was positive for Tpr, 153 had an *Arhgef11* mutation (c.2090C>T, p. Pro697Leu), and both 153PT and 153LMT were positive for Arhgef11 (Fig. 2d).

The most notable finding was that mouse 153 had distinct morphologies in its PT and LMT (Fig. 2c). However, WES identified two putative drivers, *Arhgef11* and *Plekha5*, in the 153LMT but failed to identify *Plekha5* in the 153PT, suggesting that 153PT and 153LMT have a common origin. Consistently, IHC revealed that 153PT and 153LMT were positive for Arhgef11, suggesting that *Arhgef11* might serve as a common driver (oncogene) in these two tumors (Fig. 2d). We noted that in 153PT, an aggressive invasion front that was positive for Vimentin was observed (Fig. 2c) and eventually, these aggressive tumor cells invaded into the liver, overgrowing and destroying hepatocytes (Fig. 2c). These observations suggest that 153LMT might be derived from a subpopulation of cells in 153PT; however, some driver genes in this subpopulation of 153PT might have failed to be detected due to the small number of cells or tumor heterogeneity in spatial localization. If this was the case, single-cell sequencing may be able to identify them.

**scWES of mammary and tumor tissues with *Brca1* deficiency**. Diverse VAFs of driver genes in our bulk sequencing suggested that varying degrees of tumor purity and intratumor heterogeneity exist within the tumors. In this case, some driver mutations may be hidden in the bulk sequencing. To further identify the driver mutations that might be missed by bulk sequencing, understand how tumors evolve from premalignant mammary glands upon loss of *Brca1* and find early lesions that may serve as driver mutations for tumor initiation, progression, and metastasis, we decided to conduct scWES on samples from different stages prior to tumor formation (4-month-old virgin and tumor-adjacent mammary tissues), as well as on PT and metastatic tumor samples (Fig. 3a; Supplementary Data 1). WES of bulk and single cells from the *BRCA1*-deficient patient-derived xenograft (PDX) models were also conducted (Supplementary Data 1). It is known that Cre-loxP mediated recombination is usually incomplete[32]. Using this feature, it is possible to obtain and distinguish *Brca1*-wild-type (*BRCA1*-WT) and *Brca1*-mutant (*BRCA1*-MT) single cells from the same mammary tissue to filter out germline mutations and technique errors introduced by single-cell sequencing. To achieve this, we intercrossed *mT/mG* mice[33] with *Brca1$^{co/co}$;MMTV-Cre* mice to generate *Brca1$^{co/co}$;MMTV-Cre;mT/mG* (*Brca1$^{MKO}$-mT/mG*) mice, which not only helped to trace Cre-loxP-mediated deletion of *Brca1* exon 11 but also enabled us to obtain *Brca1*-WT and *BRCA1*-MT single cells from the same mammary gland (Fig. 3b; Supplementary Fig. 3a, b, c, d).

The tumors that developed in the *Brca1$^{MKO}$-mT/mG* mice were largely green fluorescent protein (GFP) positive (Fig. 3c; Supplementary Fig. 3a), indicating that they were derived from *Brca1*-MT mammary epithelial cells, which was also confirmed by PCR genotyping (Supplementary Fig. 3c) and next-generation sequencing (NGS) (Supplementary Fig. 3d). The data also indicated that tumors had a high heterogeneity, with mixed stromal cells seen within a single tumor (Fig. 3d), reflecting

heterotypic interactions between cancer cells and noncancer cells in the *Brca1* tumor microenvironment. To identify the real driver genes and study their role in driving tumorigenesis and metastasis, we captured single cells using the C1 platform.

To ensure that DNA isolated from these single cells was of good quality and representative of the whole genome, we developed primers for each chromosome of mice and humans and conducted PCR analysis (Supplementary Fig. 3e, f). The cells that were positive for all chromosomes with the correct genotype were used for further analysis. From the 934 single cells we captured, WES was conducted on 135 single cells that met the above criteria (Supplementary Data 1). In these 135 single cells, 98 single cells from four different tumor development stages of *Brca1$^{MKO}$-mT/mG* mice and 37 single cells from PDX models were used (Supplementary Data 1). A mean depth of 217X in single cells was obtained (Supplementary Data 1). To avoid the false-positive results, we called somatic SNVs using individual-paired kidney tissues as a control and selected the SNVs that were present in three or more single cells[34].

We identified a total number of somatic SNVs per cell or bulk ranging from 17 to 104 in *Brca1$^{MKO}$-mT/mG* mice (Supplementary Fig. 3g; Supplementary Data 2, 5) with different types of mutations, including synonymous, missense, splice site, stop gained, and start lost; missense was a predominant mutation type (Fig. 3e). The SNVs of critical genes for further investigation in this study were validated using PCR-Sanger sequencing (Supplementary Fig. 3i; Supplementary Data 4). The base substitution spectrum revealed that C>T/G>A was the predominant change in each single cell from the different tumor developmental stages analyzed (Supplementary Fig. 3h), and paired bulk sequencing showed a very similar pattern (Supplementary Fig. 1c and Supplementary Fig. 3h), which provides evidence to prove that our single-cell somatic SNV calling was reliable after filtering using stringent criteria. As mentioned earlier, all cells at different tumor development stages were strictly selected using multiple criteria, and we then determined whether the number of cells sequenced was high enough to provide sufficient genetic information that changed in different tumor developmental stages. The upper limit of detection for somatic SNVs captured from single cells was evaluated by increasing the number of SNVs with an increasing number of cells from each mouse[25]. As expected, the number of SNVs increased with increasing cell numbers and reached a plateau of cumulative SNVs at a specific number of cells in virgin mouse, mouse 476, and mouse 153, respectively (Fig. 3f). These data suggest that we had captured the majority of the somatic SNVs within each mouse; thus, the sample sizes are considered sufficient for identifying driver mutations and the main clonal architecture of these malignancies. However, due to the lack of matched normal control from the same patients, the reliable somatic SNVs results of human single cells from PDX models cannot be obtained and discussed.

**Driver mutations and clonality of *Brca1*-deficient tumors**. To identify putative somatic driver mutations with a functional impact on tumor initiation, progression, and metastasis, we next focused on the nSNVs. We identified a range of 1–15 nSNVs per cell (Fig. 4a). Our analyses indicated that while some nSNVs could be found in both bulk and single cells, many others could only be identified in single cells, suggesting the greater sensitivity of single-cell sequencing for identifying driver mutations compared to bulk sequencing (Fig. 4b). In addition, even though some mutations were shared by many single cells, there were no identical cells in any samples, which suggests that *Brca1*-deficient mammary glands and tumors have high genomic heterogeneity

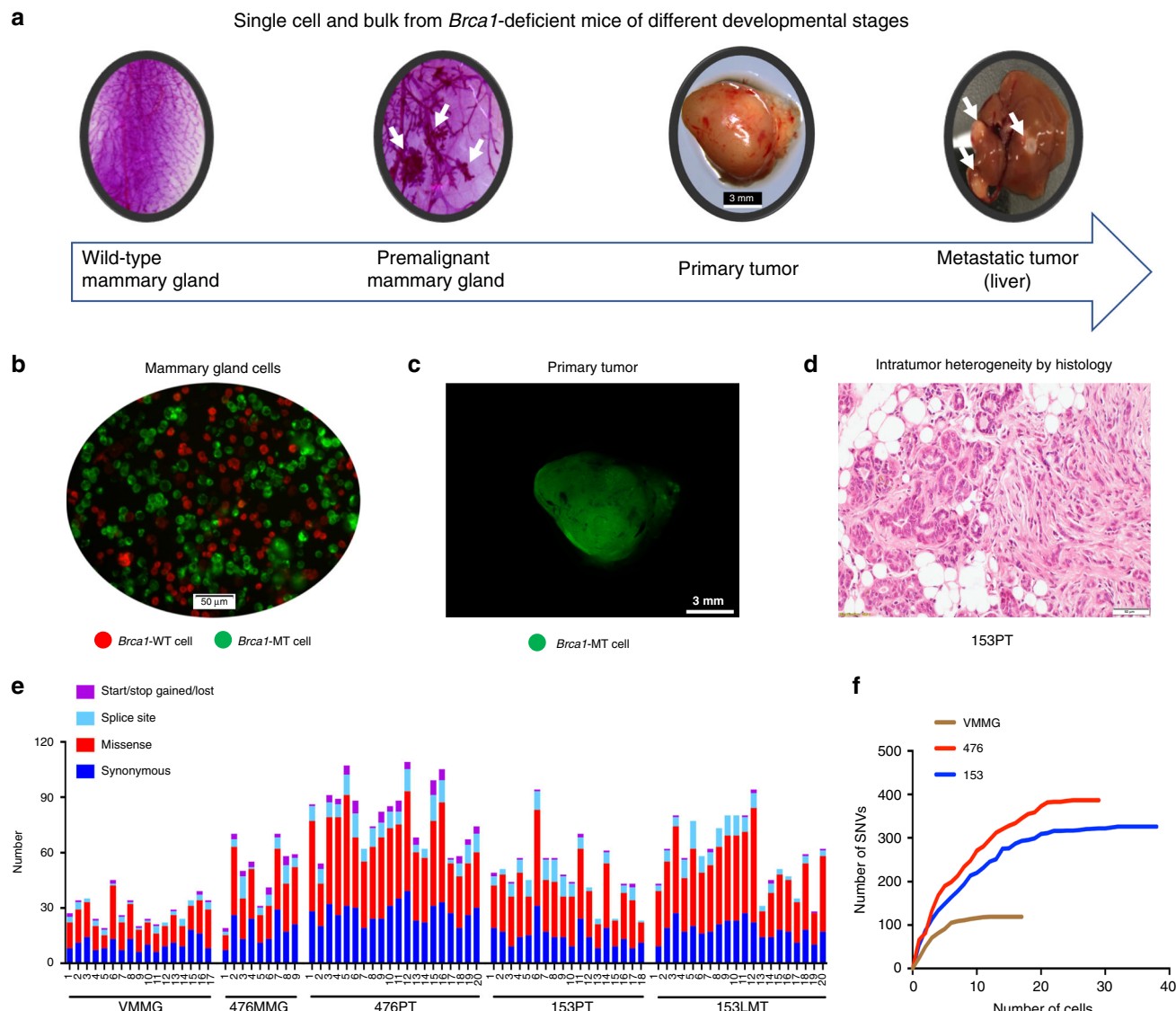

**Fig. 3 Somatic SNV spectrum of mammary and tumor tissues with *Brca1* deficiency revealed by scWES. a** The single cells and paired bulk tissues from different tumor developmental stages of *Brca1+/+;MMTV-Cre;mT/mG* and *Brca1co/co;MMTV-Cre;mT/mG* mice were sequenced. Wild-type mammary gland refers to the *Brca1*-wild-type mammary gland from 4-month-old mice with normal morphogenesis of mammary ductal tree. The premalignant mammary gland indicates the *Brca1*-deficient mammary gland with abnormal morphogenesis of mammary ductal tree. The arrows of the second panel indicate premalignant mammary gland hyperplastic foci. The arrows of the fourth panel indicate metastatic tumors buried in the liver. Scale bar, 3 mm. **b** Representative *Brca1*-WT and *Brca1*-MT mammary gland cells from *Brca1*-deficient mammary gland tissue. GFP (green) indicates *Brca1*-MT cells, Tomato (red) indicates *Brca1*-WT cells. Scale bar, 50 μm. **c** Representative *Brca1*-deficient primary breast tumor from *Brca1co/co;MMTV-Cre;mT/mG* mouse. GFP indicates *Brca1*-MT cells. Scale bar, 3 mm. **d** Representative H&E staining of the primary tumor from mouse 153 showing intratumor heterogeneity. Scale bar, 50 μm. **e** Quantification of exonic mutations according to the mutation type of *Brca1*-MT cells from different tumor developmental stages. VMMG, virgin mutant mammary gland cells (i.e., early stage of premalignant mammary gland cells from 4-month-old mouse); 476MMG, 476 mutant mammary gland cells (i.e., late stage of premalignant mammary gland cells from 11-month-old tumor-bearing mouse #476); 476PT, 476 primary tumor cells (i.e., malignant primary tumor cells from 11-month-old tumor-bearing mouse #476); 153PT, primary tumor cells (i.e., malignant primary tumor cells from 11-month-old tumor-bearing mouse #153); 153LMT, liver metastatic tumor cells (i.e., malignant metastatic tumor cells from 11-month-old tumor-bearing mouse #153). **f** Saturation curve of SNVs of single cells in each mouse.

(Supplementary Fig. 4a–c). Consistent with our previous bulk tumor analysis, nSNVs revealed by scWES showed a mouse-specific pattern (Fig. 4b). Notably, tumor-adjacent mammary cells and corresponding PT cells shared similar nSNVs (Fig. 4b; Supplementary Fig. 4b), suggesting that some of these mutations might be responsible for tumor initiation and be able to serve as early markers for cancer. In addition, PT cells and metastatic tumor cells from the same mouse shared similar nSNVs (Fig. 4b; Supplementary Fig. 4c), indicating that PT and metastatic tumor

come from the same origin and that these mutations affect the genes that not only drive tumor initiation but also drive metastasis. Notably, one common gene, *Gbp4*, was identified in three tissues of two different mice (Fig. 4b).

To identify highly mutated genes, we listed genes that were mutated among all single cells analyzed (Fig. 4c). The top-ranked genes included *Arhgef11*, *Plekha5*, *Scube3*, *Akt1s1*, *Psmb3*, and *Gbp4*. Besides the effect of a mutation, the function of a gene was taken into account for a gene being a potential driver for each cell.

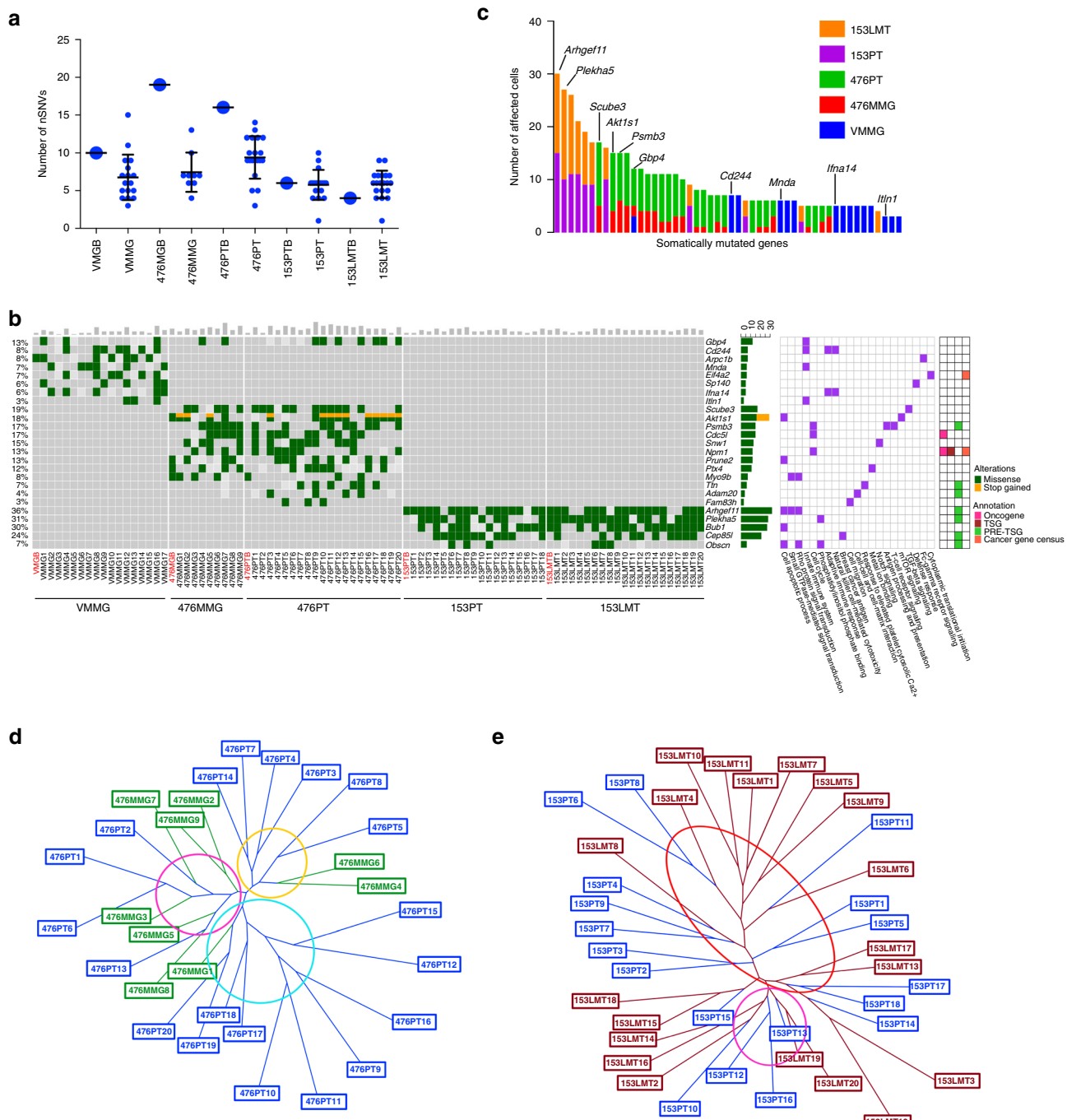

**Fig. 4 Driver mutations and clonality of *Brca1*-deficient breast tumors revealed by scWES. a** Number of nonsynonymous SNVs of human homologous genes in single cells and paired bulk samples from different tumor developmental stages. A big dot indicates a bulk sample, a small dot indicates a single cell (VMGB, $n = 1$ bulk tissue; VMMG, $n = 17$ cells; 476MGB, $n = 1$ bulk tissue; 476MMG, $n = 9$ cells; 476PTB, $n = 1$ bulk tissue; 476PT, $n = 20$ cells; 153PTB, $n = 1$ bulk tissue; 153PT, $n = 18$ cells; 153LMTB, $n = 1$ bulk tissue; 153LMT, $n = 20$ cells. Error bars represent SD). **b** Oncoplot summarizing recurrently altered genes in single cells and paired bulk samples according to the developmental stages. A gray rectangle indicates the wild-type status of the indicated site, a light gray rectangle indicates no coverage of the indicated site. **c** Incidence plot of frequently mutated genes (≥3 affected cells) detected in the series ($n = 50$ genes; $n = 84$ cells). **d** The neighbor-joining phylogenetic tree showing the clonality of 476MG and 476PT. The subclones were circled and determined by at least more than four cells distributed in a trunk. **e** The neighbor-joining phylogenetic tree showing the clonality of 153PT and 153LMT. The subclones were circled and determined by at least more than four cells distributed in a trunk.

Some of these genes, such as *Scube3*[35] and *Akt1s1* (also known as *Rpas40*)[36], were reported to be involved in tumorigenesis, while the roles of some others have not been reported. Interestingly, *Gbp4* was mutated in the early (virgin mutant mammary gland (VMMG)) and late (476MMG) stages of premalignant transformation in the mammary gland, as well as in the malignant primary tumor (476PT) (Fig. 4b, c). Gbp4 was reported to be involved in the innate immune system and induced by *IFN-γ*[37]. Gbp4 was able to negatively regulate IFN-α by targeting IRF7 in the antiviral response[38]. In addition to *Gbp4*, mutations of several putative cancer driver genes, such as *Scube3*[35], *Akt1s1*[36], and *Psmb3*[39,40], that affect tumorigenesis were identified in both

mutant mammary glands and PTs of mouse 476 by single-cell sequencing (Fig. 4b, c; Supplementary Fig. 4b), but some of these mutations were not detected by bulk sequencing, suggesting that cell-intrinsic factors and extrinsic microenvironmental factors both are essential for tumor initiation and progression.

*Arhgef11* was the most frequently mutated gene in mouse 153 (153PT: 15/18, 83%; 153LMT: 15/20, 75%) (Fig. 4b, c; Supplementary Fig. 4c). This gene encodes a protein that contains a pleckstrin homology (PH) domain and is a guanine nucleotide exchange factor that upregulates Rho GTPase, which plays an essential role in cell motility, migration, and growth of invasive breast cancer cells[41,42]. Notably, *Plekha5* was also mutated in PT cells (153PT: 10/18, 55.5%) and most often mutated in metastatic tumor cells (153LMT: 17/20, 85%), as revealed by single-cell sequencing (Fig. 4b, c; Supplementary Fig. 4c), but it failed to be detected in PT bulk sequencing (Fig. 1c; Table 1). A conceivable explanation is that cells carrying this mutation represent a minor population in the PT, and the mutation could be hidden in background noise and filtered away by bulk sequencing. However, in the case of single-cell sequencing, it was easily identified in the mutated cells. *Plekha5* encodes a protein containing a PH domain, but the role of *Plekha5* in tumor formation and metastasis remains elusive. Notably, analysis of publicly available sequencing data of 2606 breast cancer patients[43–46] with cBioportal[47] (http://www.cbioportal.org) also identified mutations of *Arhgef11* and *Plekha5* in 2.01% and 0.5% of patients, respectively. Intriguingly, a tendency for the co-occurrence of mutations in *Arhgef11* and *Plekha5* was observed in our single-cell data (Supplementary Fig. 4d), as well as in analyzed human breast cancer data (Supplementary Fig. 4e).

To further understand the intratumor heterogeneity and tumor clonal structure, we performed a phylogenetic analysis to assess tumor clonality using a modified neighbor-joining method[48]. Our results revealed that: (1) 476PT consisted of at least three major subclones and might originate from three different *Brca1*-MT mammary gland cells (Fig. 4d). This high heterogeneity is consistent with the various mutations with low VAFs observed in the single-cell and bulk sequencing (Table 1; Fig. 1c; Supplementary Fig. 4b). (2) 153PT was more homogeneous driven by *Arhgef11* for tumor initiation (Fig. 4e; Supplementary Fig. 4c, f). Later, a small population of cells with the enriched mutation in *Plekha5* became the dominant clones in 153LMT (Supplementary Fig. 4c, f). Of note, 65% (13/20) of 153LMT cells and 44% (8/18) of 153PT cells contained both the *Arhgef11* mutation and the *Plekha5* mutation, highlighting the power of single-cell technology in predicting possible collaborative actions for tumor evolution and tumor metastasis.

**CNV driving forces and evolution in *Brca1*-deficient tumors.** Bulk CNV results provided an average number of CNVs with low resolution in a certain tumor, which is affected by tumor purity and intratumor heterogeneity. To better understand CNVs within tumors and identify driver events, we decided to analyze CNVs from scWES data. CNV calling based on scWES is limited by sequenced regions from WES and challenged by algorithms for single-cell CNV calling. To overcome these difficulties, we reasoned that averaging the RPKM value of a large number of genomically adjacent genes would provide a gene-specific average RPKM and reflect chromosomal CNVs[49]. We attempted to call CNVs from our scWES data using this principle and investigated the pre-malignant changes in *Brca1*-MT mammary epithelial cells and tumor cells. To validate single-cell CNV calling, we first compared the CNV profile of single cells with the CNV profile of paired bulk samples called using a popular tool, CNV kit[50]. For each mutant mammary gland or tumor, a common pattern of aneuploidy was

revealed by both the single cells and the bulk (Supplementary Fig. 5a–e), such as amplification of chromosome 11 in 476PT and amplification of chromosome 1 and 16 in 153PT and 153LMT. These data suggest that the way we called single-cell CNVs using scWES data is reliable. More obvious amplifications or deletions observed in single cells demonstrate the power of single-cell sequencing for providing higher resolution results (Supplementary Fig. 5a–e). Interestingly, a common amplification or deletion pattern of CNVs was only found in the tumor stage (476PT, 153PT, and 153LMT) but not in the premalignant stage (VMMG and 476MMG) (Supplementary Fig. 5a–e), which might indicate that *Brca1* deficiency results in random genetic events by causing genome instability in the premalignant stage, whereas driver genetic events accumulate and become dominant later in tumor stage.

To further understand how tumors evolve from premalignant mammary glands to PTs and metastases due to copy number changes upon the loss of *Brca1*, we analyzed all cells based on the tumor developmental stages in mice in the following order: VWMG, VMMG, 476WMG, 476MMG, 476PT, 153PT, and 153LMT (Fig. 5a; Supplementary Data 3). Interestingly, we found that (1) an amplification or deletion pattern was only present in tumor cells but not in normal and premalignant *Brca1*-deficient mammary gland cells; (2) different tumors from different mice had different CNV profiles; (3) metastatic tumor shared very similar CNV profile with paired PT. When we further examined CNV profile in detail, we found that chromosome 11 was obviously amplified in 476PT cells but not in paired mutant mammary gland cells (Fig. 5a; Supplementary Fig. 5h) and amplification of chromosome 1q, 6p, 16q, and deletion of chromosome 4q, 14 were observed in both 153PT and 153LMT (Fig. 5a; Supplementary Fig. 5i). Similar to bulk tumor CNV data, single tumor cells of mouse 153 showed a much different pattern from those of mouse 476 (Fig. 5a), suggesting different driving forces in different individual tumors. Similar amplification and deletion patterns were shown in PT single cells and metastatic tumor single cells in mouse 153 (Fig. 5a), indicating that they came from the same origin. Combining these results with the single-cell SNV results, we found that (1) SNVs occurred earlier in the premalignant stage, which might play an important role in tumor initiation (Fig. 5b); (2) CNVs were acquired later in a burst, which characterized the malignancy and promoted tumor formation and progression (Fig. 5b); and (3) metastatic tumors originate from PTs with increased frequency of co-occurrence of two SNV mutations (*Arhgef11* and *Plekha5*) from 44 to 65% and CNVs within the same single cells, indicating the contribution of CNVs to the driving force for tumor metastasis (Fig. 5c). Based on this analysis, we constructed evolution models for *Brca1*-deficient tumors (Fig. 5b, c), showing that tumors initiate from cells with SNVs affecting potential driver genes in premalignant stage and progress via CNVs acquired in chromosome regions with many cancer driver genes.

Next, we investigated the pathways and genes affected by copy number changes and performed DAVID–KEGG pathway analysis[51] of genes in the amplified and deleted regions. We found that many cancer-driving pathways were involved, including the pathways in cancers, the MAPK signaling pathway, the PI3K–Akt signaling pathway, and the focal adhesion pathway, which contain many well-known cancer driver genes (Supplementary Fig. 5j–l). Strikingly, a well-known cancer-driving gene, *Ppm1d*, which encodes a *p53*-induced phosphatase, was found in 476PT cells with chromosome 11 amplification (Fig. 5a; Supplementary Fig. 5j). Ppm1d negatively regulates p53 through attenuation of p38 MAPK activity, therefore inactivating p53 and downregulating p53-mediated transcription and apoptosis[52]. Since no *p53* mutations were detected in mouse 476 (*Brca1^co/co^; MMTV-Cre;mT/mG*), we believed that the amplification of

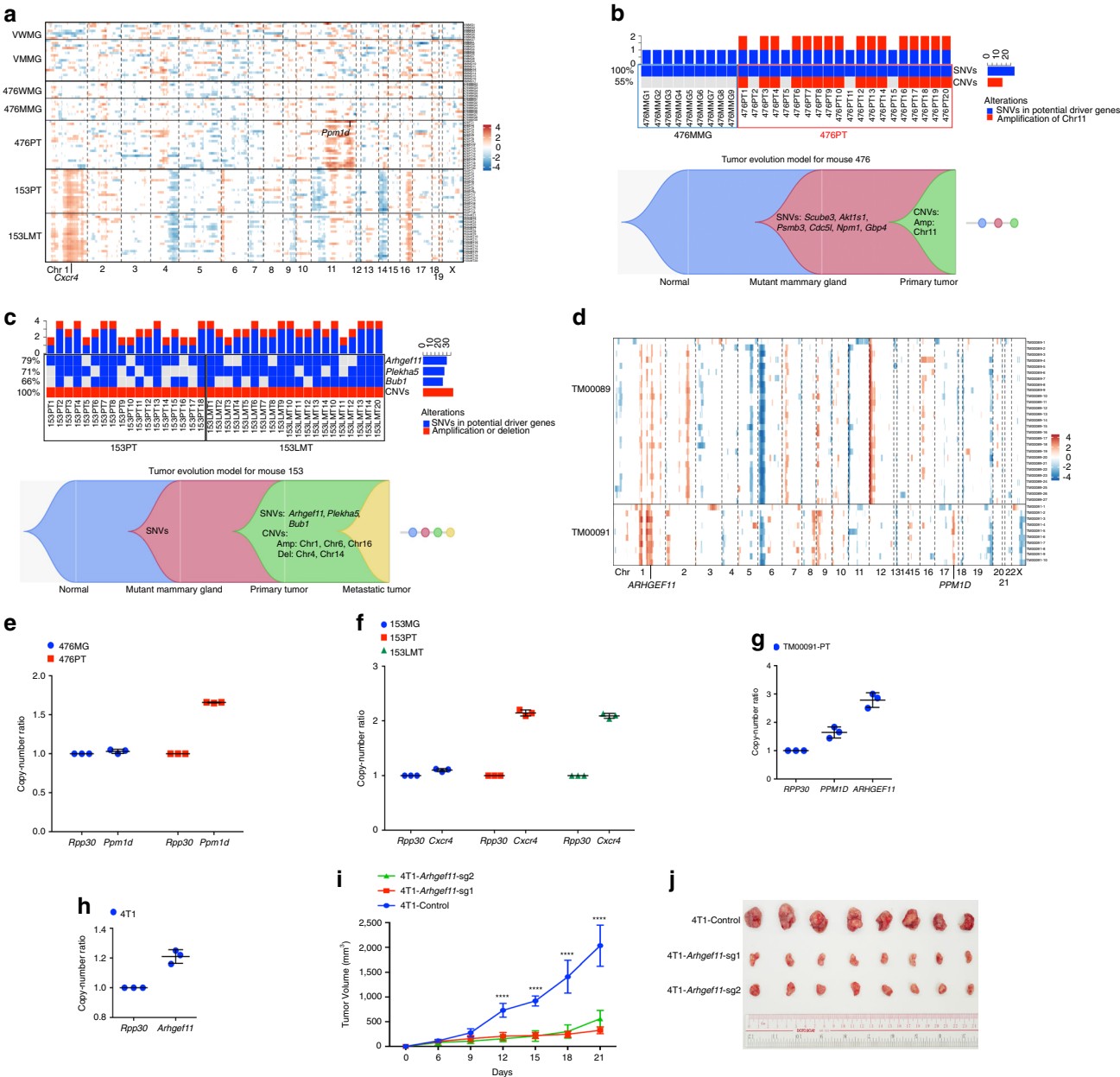

**Fig. 5 Driver events of CNVs and tumor evolution of *Brca1*-deficient breast tumor revealed by scWES. a** DNA copy-number profiles based on the developmental stages of single cells from mice (*n* = 98 cells). VWMG-virgin wild-type mammary gland cells from 4-month-old mouse; 476WMG-476 wild-type mammary gland cell from 11-month-old tumor-bearing mouse #476. **b** Top: oncoplot summarizing SNVs in potential driver genes (*Scube3, Akt1s1, Psmb3, Cdc5l, Npm1*, etc.) (Fig. 4b; Supplementary Fig. 4b) and potential driver CNVs (amplification of chromosome 11) (**a**) of mouse 476. Bottom: tumor evolution model of mouse 476 based on potential driver SNV and CNV results. **c** Top: oncoplot summarizing SNVs in potential driver genes (*Arhegf11, Plekha5, Bub1*, etc.) (Fig. 4b; Supplementary Fig. 4c) and potential driver CNVs (amplification of chromosome 1, 6, 16, and deletion of chromosome 4, 14) (**a**) of mouse 153. Bottom: tumor evolution model of mouse 153 based on potential driver SNV and CNV results. **d** DNA copy number profiles of single cells from the *BRCA1*-MT PDX models (*n* = 37 cells). TM00089, which carries a frameshift variant: *BRCA1*-V757fs; and TM00091, which carries a common pathogenic missense variant C61G in *BRCA1* RING domain. **e–h** ddPCR analysis of amplification of chromosome 11 (*Ppm1d*) in 476MG and 476PT (**e**), amplification of chromosome 1 (*Cxcr4*) in 153MG and 153PT (**f**), amplification of chromosome 1 (*ARHGEF11*) and chromosome 17 (*PPM1D*) in TM00091 (**g**), amplification of *Arhgef11* in 4T1 cell line (**h**). *Rpp30/RPP30* was used as a reference gene for normalization. Three technical replicates were performed. **i** Measurement of tumor volume after implantation of 4T1 Control and 4T1-*Arhgef11*-KO cells into mammary fat pad (4 × 10⁶ tumor cells per injection, *n* = 8 mice per group). Data are reported as the mean ± SD with two-way ANOVA analysis with Dunnett's test, ****$P < 0.0001$. **j** Image of tumors in (**i**) at the endpoint of measurement. Error bars represent SD.

*Ppm1d* might play a role in suppressing p53 and driving tumor formation in this scenario. The oncogene *Cxcr4* was found in the amplified region of chromosomes 1 in 153PT and 153LMT cells (Fig. 5a; Supplementary Fig. 5k). *Cxcr4* has been reported to be involved in many important processes surrounding tumor

formation and metastasis, including cell survival, proliferation, adhesion, migration, and invasion[53]. Besides, several TSGs, including *E2f2, Wnt5a*, and *Bmp8b*, etc. were found in the deleted regions of chromosomes 4 and 14 in 153PT and 153LMT cells (Fig. 5a; Supplementary Fig. 5l). These data are

consistent with the observations of tumor heterogeneity evidenced by the many different mutations seen in different cells and the enrichment of subpopulations with potent driver mutations promoting tumor growth and metastasis.

To determine whether there were similar CNV events in both human and mouse breast cancers, we then used the same approach to study CNVs from two human *BRCA1*-deficient (TM00089 and TM00091) xenograft breast tumors by single-cell and bulk WES. The data showed similar CNV profiles in single-cell and bulk samples (Supplementary Fig. 5f, g), with some differences. Similar and different CNV patterns were observed in TM00089 and TM00091 (Fig. 5d). Amplification of chromosome 1 and deletion of chromosome 6p was observed in both TM00089 and TM00091, with the latter had greater amplification of chromosome 1 (Fig. 5d). Amplification of chromosome 12p was specifically found in TM00089, and amplification of chromosome 17q was specifically present in TM00091 (Fig. 5d). Interestingly, *ARHGEF11*, which was found highly mutated in 153PT and 153LMT cells (Fig. 4b, c; Supplementary Fig. 4c) with strong protein signaling (Fig. 2d) in the PT and LMT, was identified in the amplified region of chromosome 1 of TM00091. In addition, *PPM1D*, a gene amplified and described in 476PT cells, was identified in the amplified region of chromosome 17q of TM00091. Of note, this region is a region that is homologous to the region of chromosome 11 previously showing amplification in 476PT cells (Fig. 5a) and is most commonly amplified in breast cancer[54]. DAVID–KEGG pathway analysis of genes in the amplified region of TM00091 revealed that the pathways in cancer, the PI3K–AKT signaling pathway, the MAPK signaling pathway, etc. were involved in this *BRCA1*-deficient tumor (Supplementary Fig. 5m). Moreover, we extended the analysis of copy number amplification of *ARHGEF11* and *PPM1D* to a cohort of breast tumors (4563 cases)[43–46,55,56] using cBioportal[47] (http://www.cbioportal.org) and revealed amplification of *ARHGEF11* and *PPM1D* in 14.27% and 9.21% of patients, respectively. These results revealed that amplification of chromosome 17 in humans (equivalent to chromosome 11 in mice), which contains driver genes (*PPM1D*, etc.) for breast cancer, might be a hot-spot event at least in a portion of *BRCA1*-deficient breast cancers. In addition, other individual-specific driver events also play important roles in tumor initiation, progression, and metastasis. To further validate the CNV calling in single cells, we performed a highly sensitive and quantitative ddPCR assay, and we confirmed the copy-number gain of *Ppm1d* (Fig. 5e) and *Cxcr4* (Fig. 5f) in mouse tumors, but not in mammary glands of the same mouse, and the gain of *ARHGEF11* and *PPMD1* in TM00091 tumor from human (Fig. 5g) with *Rpp30/RPP30* as a reference.

To determine if CNVs affect the protein level in these samples, we performed IHC on the tissues harboring corresponding gene amplifications to examine the protein levels. As expected, 476PT but not 476MG was positive for Ppm1d (Supplementary Fig. 5n), and 153PT and 153LMT were both positive for Cxcr4 (Supplementary Fig. 5o). High levels of PPM1D and ARHGEF11 were present in TM00091, but not in TM00089 (Supplementary Fig. 5p, q). Altogether, these data were consistent with our WES results and suggested that enriched CNVs (*Ppm1d/PPM1D*, *Cxcr4*, *ARHGEF11*, etc.) in tumors were driving forces for progression and metastasis in these tumors.

Finally, to understand the role of *Arhgef11* in tumor progression, we performed CNVs screen in several breast cancer cell lines using ddPCR and found copy-number gain of *Arhgef11* in 4T1 cells (Fig. 5h). Then, we knocked out *Arhgef11* in 4T1 cells using the CRISPR/Cas9 system with two different sgRNAs (Supplementary Fig. 6a, b) and implanted the knockout cells into the mammary fat pads of BALB/c mice to monitor tumor growth. The results showed much smaller tumor volume

observed in *Arhgef11*-KO groups than the control group (Fig. 5i, j), suggesting copy-number gain of *Arhgef11* is a tumor driving event and *Arhgef11* plays an oncogene role in tumor progression.

**Plekha5 is a tumor metastasis suppressor.** Our analysis implicated a role of *Plekha5* in metastasis. To understand the role of *Plekha5* in the metastatic process, we further examined the protein level of Plekha5 in 153PT and 153LMT tissues by IHC and found that Plekha5 had weak staining in the PT and much weaker staining in the LMT (Fig. 6a), suggesting that *Plekha5* might serve as a metastatic repressor.

To test its function, we knocked out *Plekha5* using the CRISPR/Cas9 system in a GFP-labeled *Brca1*-WT mouse mammary epithelial cell line with low metastatic potential (B477-GFP) (Fig. 6b; Supplementary Fig. 6c). Then, we conducted transwell migration and invasion assays using this cell line. The results showed that *Plekha5* deficiency promoted cell migration and invasion (Fig. 6c). To test its function in vivo, we implanted the knockout cells into the mammary fat pads of nude mice to monitor cell metastasis to distant organs. We observed numerous strong GFP-positive signals in the liver 1 month after implanting B477-GFP-*Plekha5*-knockout (B477-GFP-*Plekha5*-KO) cells (Fig. 6d, e). GFP-positive signals were observed in 10/10 metastases in the *Plekha5*-KO group and 2/9 metastases in the control group (Fig. 6j), which had a few GFP-positive cells. Sanger sequencing analysis of tumor DNA from GFP-positive liver cells confirmed that the GFP-positive cells in the liver were from the PT based on the presence of shared mutations (Supplementary Fig. 6d). Moreover, we examined CK14, a myoepithelial cell marker for mouse mammary glands, in frozen metastatic liver tissue sections. Positive CK14 signals were detected in metastatic liver samples (Fig. 6d), suggesting that they originated from the mammary gland. In addition, we performed *Plekha5* knockout in a GFP-labeled mouse *Brca1*-deficient mammary tumor cell line with low metastatic potential (G600-GFP) and implanted these cells into mice (Supplementary Fig. 6e). The results showed an increased incidence of metastasis in the *Plekha5*-KO group (6/8) compared to that seen in the control group (1/8) (Fig. 6j). These data provide strong evidence that *Plekha5* acts as a metastasis suppressor regardless of *Brca1* status.

To further investigate the metastatic function of mutated *PLEKHA5* in human breast cancer, we knocked out *PLEKHA5* in GFP-labeled *BRCA1*-WT MDA-MB-231 cells (231-GFP) (Fig. 6f; Supplementary Fig. 6f). Similarly, our transwell assay indicated that the knockout of this gene in human cells could also promote cell migration and invasion (Fig. 6g). We then implanted the knockout cells into the mammary fat pad of nude mice to monitor cell metastasis to distant organs and observed more metastases to the lung in the *PLEKHA5*-KO group compared to the control group (Fig. 6h–j).

Notably, the depletion of *Plekha5/PLEKHA5* in mouse and human cells had no obvious effect on tumor growth (Supplementary Fig. 6g–i). These results demonstrated that knockout of *Plekha5/PLEKHA5* could promote cell metastasis to distant organs such as the liver or lung, in both *BRCA1*-deficient and *BRCA1*-WT breast cancer in both humans and mice. To extend our findings to the human database, we analyzed *PLEKHA5* expression in a collection of 155 breast tumors with or without metastasis[57]. Consistent with our data, the expression level of *PLEKHA5* was significantly lower in the PTs with metastasis compared with the PTs without metastasis (Fig. 6k). As expected, overall survival analysis revealed that patients bearing PTs with metastasis had much worse overall survival rates in this cohort of patients (Supplementary Fig. 6j). We further determined whether there was a correlation between *PLEKHA5* expression and overall survival,

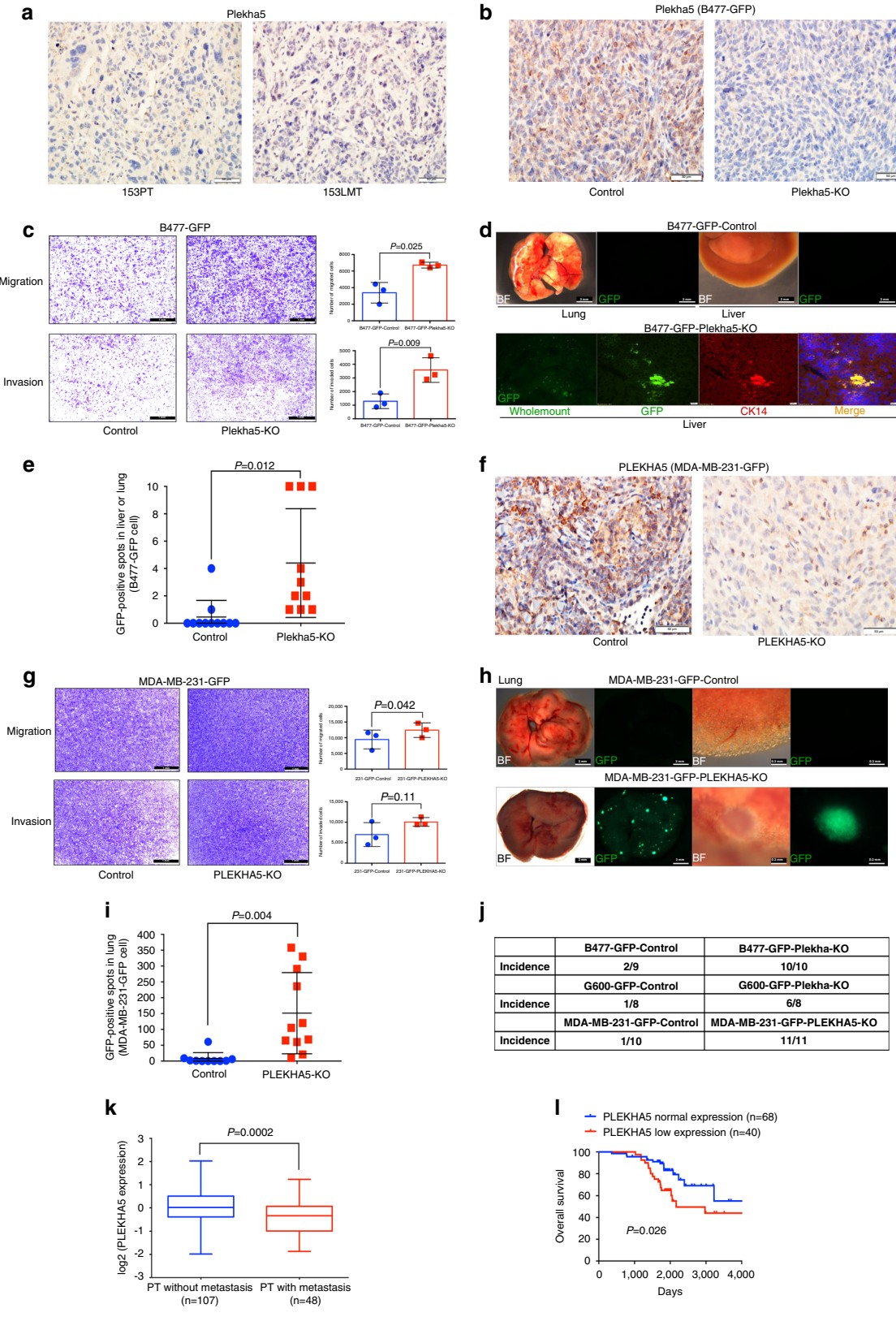

and the data revealed that low expression of *PLEKHA5* was associated with worse overall survival in this cohort of patients (Fig. 6l). Collectively, all this evidence of *PLEKH5* in human breast cancers is consistent with the data obtained from functional studies in mice, which suggests that *PLEKHA5* is a tumor metastasis suppressor of breast cancer.

## Discussion

In this study, we have performed simultaneous SNV and CNV analysis in bulk and single cells of *BRCA1*-deficient breast cancers using WES. We uncovered several notable findings that have not been clearly illustrated previously. (1) We identified a tumor evolution process through which tumors initiate from cells with

**Fig. 6 _Plekha5_ is a tumor metastasis suppressor. a** Representative IHC images of Plekha5 in 153PT and 153LMT. Scale bar, 50 μm. **b** Representative IHC images of Plekha5 in primary tumor formed by _Plekha5_-KO B477-GFP cells injected into the mammary fat pad of nude mice. Scale bar, 50 μm. **c** Transwell assays using _Plekha5_-KO B477-GFP cells. Scale bar, 1 mm ($n = 3$ replicates over three independent experiments). Data are reported as the mean ± SD. Significance determined by two-tailed Student's $t$ test. **d** Metastasis functional assay in vivo using _Plekha5_-KO B477-GFP cells. Top: wholemount lungs or livers from control mice. Bottom: wholemount or frozen section of livers from mice implanted with _Plekha5_-KO cells. Left four, merged image of left two and three with 4′,6-diamidino-2-phenylindole (DAPI). Representative images were shown. Scale bar: wholemount, 3 mm; frozen section: 20 μm. BF, bright field. **e** Quantification of GFP signals in organs of mice ($n$ (control group) = 9 mice, $n$ (Plekha5-KO group) = 10 mice). Count the number of more than 10 as 10. Significance determined by two-tailed Student's $t$ test. **f** Representative IHC images of PLEKHA5 in primary tumor formed by _PLEKHA5_-KO MDA-MB-231-GFP cells injected into the mammary fat pad of nude mice. Scale bar: 50 μm. **g** Transwell assays for _PLEKHA5_-KO MDA-MB-231-GFP cells. Scale bar: 1 mm ($n = 3$ replicates over three independent experiments). Data are reported as the mean ± SD. Significance determined by two-tailed Student's $t$ test. **h** Metastasis functional assay in vivo using _PLEKHA5_-KO MDA-MB-231-GFP cells. Top: wholemount (scale bar, 3 mm) and enlarged (scale bar, 0.3 mm) images of lungs from control mice. Bottom: wholemount (scale bar, 3 mm) and enlarged (Scale bar, 0.3 mm) images of lungs from mice implanted with _PLEKHA5_-KO cells. **i** Quantification of GFP signals in organs of mice ($n$ (control) = 10 mice, $n$ (PLEKHA5-KO) = 11 mice). Significance determined by two-tailed Student's $t$ test. **j** Metastasis incidence of mice after implanted with different _Plekha5/PLEKHA5_-KO mouse or human cell lines. **k** Expression of _PLEKHA5_ in the primary tumors without metastasis ($n = 107$ tumors) and the primary tumors with metastasis ($n = 48$ tumors). Significance determined by two-tailed Student's $t$ test. Box-and-whisker plots: center line, median; box limits, upper and lower quartiles; whiskers, minima to maxima. **l** Kaplan–Meier curves showing the correlation between _PLEKHA5_ expression and overall survival. Significance determined by a two-sided the log-rank test (Mantel–Cox). Error bars represent SD.

---

SNVs affecting putative driver genes at premalignant stages and malignantly progress later via CNVs acquired in chromosomal regions with many cancer driver genes. (2) These are random events that hit many putative cancer drivers besides _p53_ to generate unique genetic and pathological features for each tumor. (3) Cancer drivers or metastasis drivers, when they are present at low frequency or occur at premalignant stages, could be missed by bulk DNA sequencing due to high heterogeneity but can be identified by analyzing only a small number of single cells. (4) One of the examples is the _Plekha5_ gene, the mutation of which is present in a small population of cells in the PT but became dominant in LMT. (5) Our scWES results combined with CRISPR-Cas9-mediated knockout studies provide solid evidence that _Plekha5_ does not affect PT growth; however, it is a metastasis suppressor, whose deficiency promotes cancer metastasis to the liver and/or lung.

Mutation of _P53_ is frequently identified in human breast cancers with germline mutations of _BRCA1_[44]. By analyzing the WSI dataset, we found that 81% of patients with _BRCA1_ germline mutations carry _P53_ mutations (Fig. 1d). In a mouse model carrying a mammary-specific deletion of the full-length _Brca1_, we found that mammary tumors developed after a long period of latency and that the loss of _p53_ markedly accelerates tumorigenesis[14]. We also wondered whether any other factors besides _p53_ are involved in _Brca1_-associated tumorigenesis. At stages when genomic sequencing is not widely affordable, we had used a genetic approach in which we bred mutant mice carrying mutations of some genes with _Brca1_-mutant mice. Although the process was both time-consuming and labor-intensive, these studies revealed critical roles of _p53_, _Atm_, _Chk2_, and _53BP1_ in various aspects of _Brca1_-associated cancer formation[15–19]. Recently, Liu et al. also found that some other putative drivers, including CNVs of _Met_, _Myc_, _Rb1_, as well as gene fusions of MAPK and/or PI3K signaling, are involved in _Brca1_-associated cancer by analyzing transcriptional and CNV profiles[27]. Furthermore, Annunziato et al. revealed more driver genes, such as _Myc_, _Met_, or combinations of _Myc_ with _Pten_, _Rb1_, _Mcl1_, to be involved in _Brca1_-associated cancer formation by genomic analyses and functional assays[58]. Thus, _Brca1_-associated tumorigenesis could potentially be affected by many tumor suppressors and oncogenes.

Using scWES and bulk WES, we have identified some other putative tumor suppressors and oncogenes in addition to _p53_. Our analysis revealed that _p53_ is indeed the most commonly and frequently mutated gene, as shown previously[44]; in these collections of _Brca1_-associated cancer, several points deserve discussion. Frameshift mutations in _p53_ were identified in 2/7 _Brca1_$^{co/co}$;_MMTV-Cre_ tumors and the time to the first detection of tumors with or without _p53_ mutations was not significantly different (Table 1). For example, mouse 1949 had both _p53_ mutant and wild-type tumors, which originated independently at approximately the same time. In _Brca1_$^{co/co}$;_p53_$^{+/co}$;_WAP-Cre_ tumors, although _p53_ exon 5–6 was lost in all samples due to Cre-mediated deletion, which markedly accelerated tumorigenesis (Supplementary Fig. 1a), 11/16 tumors had LOH on the remaining wild-type allele of _p53_. The time to first tumor detection between the _p53_ LOH group and the group without LOH was not significantly different (Table 1). In addition, even in the tumors with _p53_ mutations, mutations of other TSGs or oncogenes were also found, suggesting that the loss of _p53_ is not sufficient, but necessary to cause full malignant transformation. These genes include _Rasal1_[59], which plays a tumor suppressor role in thyroid cancer, and _Nras_[60], which is a famous oncogenic gene in cancers. Collectively, these data indicate that targeting _p53_ or the _p53_ pathway may be a strategy for the therapy as well as the prevention of breast cancer caused by _BRCA1_ germline mutation. A precision medicine strategy to target different drivers of each individual is highly recommended in the treatment of _BRCA1_-deficient breast cancer, especially for those patients who do not respond to PARPi[10–12].

Both SNVs and CNVs were detected by many previous genomics studies on _BRCA1_-associated breast cancers[27,44,61], yet it remains unclear whether both driver events can occur in the same cell at premalignant stages and how they work. These issues can be best addressed by conducting DNA sequencing on single cells; however, single-cell CNV analysis is limited and challenged by the use of conventional bulk CNV calling algorithms. In our study, we learned from CNV analysis in a single-cell RNA sequencing study[49] and obtained reliable CNV results from scWES data. In addition, the use of the _Brca1_$^{co/co}$;_MMTV-Cre_; _mT/mG_ mouse model allowed us to obtain both _Brca1_-WT and _Brca1_-MT cells at the premalignant stage due to the incomplete recombination of the Cre-loxP system, which is a common feature of this system[32]. Taking these advantages, we analyzed both SNVs and CNVs in single cells at the same time, which enabled us to make a finding that driver CNVs were not detected in premalignant stages (i.e., virgin and tumor-adjacent tissues), although they were prominent in cancers. Loss of _BRCA1_ is known to cause genome instability[1,7] because of the essential role of _BRCA1_ in homologous recombination-directed DNA damage

repair[62]. It is therefore predicted that *BRCA1* deficiency should be able to cause both SNVs and CNVs; however, deletion of a fragment of a chromosome (CNVs) might impair the functions of all genes within the deleted fragment and cause lethality, whereas SNVs will certainly yield a much milder impact on the viability of cells. Thus, despite *Brca1* deficiency in mammary tissues resulting in apoptosis[14], cells with SNVs should have a better opportunity to survive than cells with CNVs. Somatic SNVs in known cancer genes (*P53*, *NOTCH1*, etc.) were previously found in normal tissues; however, the mutation burden (3.5 mutations per sample) in each sample was relatively low, and additional transforming events were required for cancer transformation[63]. In our study, the *Brca1*-deficient premalignant mammary gland cells had a much higher mutation burden (28.5 mutations per cell in VMMG, 48 mutations per cell in 476MMG), which might generate the scenario that multiple transforming events hit one cell. In addition, *Brca1* deficiency is no doubt one of the transforming factors in this scenario. Once cells are malignantly transformed, they should have a stronger ability to tolerate the lethal effects associated with CNVs and undergo Darwinian evolution, which may benefit cell proliferation, migration, invasion, etc., to drive tumor mass formation and eventually metastasis.

*Plekha5* is mutated in PT cells and LMT cells, suggesting its role in tumor metastasis. A previous study reported that *PLEKHA5* was associated with decreased brain metastasis-free survival but not associated with other metastatic sites in melanoma patients, and knockdown of *PLEKHA5* inhibited blood–brain barrier transmigration and invasion of cells in in vitro models[64]. To further study the role of *Plekha5* in our system, we have conducted a number of experiments to investigate the role of *Plekha5* in metastasis and found that CRISPR-Cas9-mediated knockout of *Plekha5* enhanced cell migration and invasion in vitro. In vivo orthotopic implantation assays showed that knockout of *Plekha5/PLEKHA5* in multiple cell lines promoted tumor cell metastasis to other distant organs, such as the lung and liver. Thus, different from the data shown by Jilaveanu et al.[64], our study identified *Plekha5/PLEKHA5* as a tumor metastasis suppressor gene through scWES screening and validation using CRISPR-Cas9-mediated functional assays. The exact reason for this discrepancy is currently unclear; however, one previous study identified two forms of *Plekha5* mRNA and found that long form of *Plekha5* is specifically expressed in the brain and short form of *Plekha5* is ubiquitously expressed[65], which may provide some hints for this discrepancy. Besides, many other differences were involved in these two studies, such as knockdown versus knockout, in vitro versus in vivo, different contexts, etc., which deserve further investigation. Meanwhile, we also showed that *Plekha5/PLEKHA5* deficiency does not affect tumor growth; thus, we intend to believe that *Plekha5/PLEKHA5* is a metastasis suppressor in breast cancer, which subjects to further investigation. In addition, it is also important to know the potential role of *Plekha5* in normal mammary gland development, which will also be carefully studied in the near future.

## Methods

**Mouse strains**. All mouse strains were maintained in the animal facility of the Faculty of Health Science, University of Macau, according to institutional guidelines. A condition with a 12 light/12 dark cycle, temperatures of 18–23 °C with 40–60% humidity was used for housing the mice. *Brca1*$^{+/+}$;*MMTV-Cre*;*mT/mG*, *Brca1*$^{co/co}$;*MMTV-Cre* or *Brca1*$^{co/co}$;*p53*$^{+/co}$;*Wap-Cre* (*Brca1*$^{MKO}$), and *Brca1*$^{co/co}$; *MMTV-Cre*;*mT/mG* (*Brca1*$^{MKO}$-*mT/mG*) female mice in a mixed background of FVB/129SvEv/Black Swiss were generated inhouse and mice at different age (2 weeks, 2 months, 4 months, 6 months, 8 months, 11 months) were used. The *BRCA1*-MT PDX models, TM00091 and TM00089, were obtained from the Jackson Laboratory. Female BALB/c and nude mice at age of 4–6 weeks were obtained from the animal facility of the Faculty of Health Science, University of Macau. All experiments were approved by the Animal Ethics Committees of the Faculty of Health Science, University of Macau (Protocol ID: UMARE-AMEND-100).

**Cell lines**. The immortalized mouse *Brca1*-WT epithelial cell line B477 was derived from the mammary gland of *Brca1*-WT mice (*Brca1*$^{+/+}$;*p53*$^{+/−}$). The mouse *Brca1*-deficient epithelial cell line G600 was derived from the mammary gland of *Brca1*-deficient mice (*Brca1*$^{\Delta exon11/\Delta exon11}$;*p53*$^{+/−}$). The 4T1 cell line, MDA-MB-231 cell line, and the 293FT-cell line for lentiviral production were obtained from American Tissue Culture Collection (ATCC). The 4T1 cells were cultured in RPMI 1640 medium (Gibco Life Technologies) containing 10% FBS with or without 100 IU ml$^{−1}$ penicillin, and 100 µg ml$^{−1}$ streptomycin. All other cells were cultured in Dulbecco's modified Eagle medium (DMEM) (Gibco Life Technologies) containing 10% FBS with or without 100 IU ml$^{−1}$ penicillin, and 100 µg ml$^{−1}$ streptomycin.

**WSI dataset analysis**. The SNV and CNV data of human breast cancer patients with *BRCA1* germline mutations from the WSI breast invasive carcinoma dataset[21] were downloaded from the web-based International Cancer Genome Consortium (ICGC)[66] (https://dcc.icgc.org/). The SNV data were summarized and plotted into oncoplots. Well-known TSGs or oncogenes with SNVs in each tumor were considered as potential driver genes. The CNV data were plotted into a heatmap.

**TSGs/oncogenes classification**. TSGs and oncogenes were determined using three published data resources, including TSGene[28] (https://bioinfo.uth.edu/TSGene), Oncogene Database[29] (http://ongene.bioinfo-minzhao.org/) and COSMIC Cancer Gene Census[30] (https://cancer.sanger.ac.uk/cosmic). Predicted TSGs or oncogenes were determined using a published TSG/oncogene-prediction list[31].

**Single-cell suspension preparation**. Fresh mammary glands and tumor tissues were cut into small pieces. Tissues were digested using a "Digestion I" solution (DMEM/F12 (Gibco, Life Technologies, #11330-032) supplemented with 5% fetal bovine serum (FBS) (Prime) (ExCell Bio, #FSP500), 5 µg/ml insulin (Sigma-Aldrich, #I1882), 500 ng/ml hydrocortisone (Sigma-Aldrich, #H0888), 10 ng/ml EGF(Invitrogen, Thermo Fisher Scientific, #13247-051), 300 U/ml collagenase III (Worthington, #S4M7602S), and 100 U/ml hyaluronidase (Sigma-Aldrich, #H3506) for 2 h at 37 °C and a "Digestion II" solution (HBSS (Gibco, Life Technologies, #14170), 5 mg/ml Dispase II (Roche Diagnostics, # 04942078001), and 0.1 mg/ml deoxyribonuclease (Worthington Biochemical, #LS002145)) for 5 min at 37 °C, followed by 0.25% trypsin-EDTA (Gibco, Life Technologies, #25200) treatment for 1 min and red blood cell lysis buffer (Invitrogen, Thermo Fisher Scientific, #00-4333-57) treatment for 5 min at room temperature. Samples were filtered through a 40 µm cell strainer to a 5 ml polystyrene tube. Epithelial cells were enriched using the EasySep Mouse Epithelial Cell Enrichment Kit (STEM-CELL TECHNOLOGIES, #19868).

**Isolation and whole-genome amplification (WGA) of single cell**. Enriched epithelial cells from mice were isolated using the C1 Single-Cell Auto Prep System[67]. Briefly, cells were resuspended at a concentration of 150–500 cells/µl. The cell suspension was mixed with C1 Cell Suspension Reagent (Fluidigm, # 100–5319) at the recommended ratio of 3:2 immediately before loading 5 µl of this final mix on the C1 IFC. Images of captured cells were collected using a Leica DMI 4000B microscope under bright field by Surveyor V7.0.09 MT software. Single-cell DNA extraction and WGA were performed on the C1 Single-Cell Auto Prep Integrated Fluidic Circuit (IFC) following the methods described in the manufacturer's protocol (PN 100–7135, http://www.fluidigm.com/). Enriched epithelial cells from PDX models were isolated by micro pipetting[34]. Cells were captured and placed into 0.2 ml of microcentrifuge tubes with 4 µl of PBS. Single-cell DNA extraction and WGA were performed using the REPLI-g Single-Cell Kit (QIAGEN, 150345) following the methods described in the manufacturer's protocol.

**Chromosome PCR panel**. Single-cell WGA efficiency was evaluated using chromosome-specific primers. We designed 20 pairs of primers (Integrated DNA Technologies) for mouse cells and 22 pairs of primers for human cells to target 20 or 22 loci on different mouse or human chromosomes for PCR:

Primers for mouse: Chr1, *Pou2f1* (F: ACCCACCTTAGCTTGTCGTC; R: GCT CTCTGTTCACAGTCGCC), 433 bp; Chr2, *Rin2* (F: TGCACCCGGGTTTCAATC TTC; R: TCTCACTGTGGGGATGGGAAA), 275 bp; Chr3, *Acad9* (F: TGGCGG CCTGAAACTTAGTG;R: TTCAAGTCTCTGCCGCACTC), 287 bp; Chr4, *Thrap3* (F: CTCTTGGGCGTGTACTCTGG; R: GGGGTGGAGAAAGGATTGGG), 281 bp; Chr5, *Stard13* (F: GCTGTTGTCTGTGGGTGAGA; R: AGTCAAGTGG TGAGAGCAGC), 281 bp; Chr6, *Met* (F: CGGGGAGGTGCAAACTAGAAT; R: CCAGAAACACCTGCCCTTGA), 298 bp; Chr7, *Myadm* (F: ACAAGGGACCTG GAGGAGTT; R: CTTTCCCCCGAAGAAAAGTC), 189 bp; Chr8, *Kars* (F: AAG TCGACCTCGTAGGCTTG; R: CAGGGCACTGGGCATTTTGA), 460 bp; Chr9, *Mpzl3* (F: TTGAGTTGTGCACTCGGGG; R: GGGCCGTCTGCATATAAGGTT), 242 bp; Chr10, *Cfap54* (F: TGGCGTGTGTTCCTTATCGG; R: GCTCTTTGCT AGTGTCTCCG), 208 bp; Chr11, *Mpp3* (F: GTGGCAGACACACTCGGTTA; R: GCTTCTTTTGCTGGGGACAC), 264 bp; Chr12, *Adam17* (F: CTGTGCAGAAC ATGACCTACCT; R: ACCTGGCCAGATGAGTTTGTC), 239 bp; Chr13, *Calm4* (F: TGATGGCAAGATCAGCTTTG; R: TTCACCTTCCCATCTTGGTC), 231 bp; Chr14, *Otx2* (F: GAGGACACAGCAACTGGTAG; R: GGCCTGCACAGCCTTA TATC), 130 bp; Chr15, *Osr2* (F: AAATGACGCGACAGCTAACC; R: ACACAAA GACAACGCCCATC), 147 bp; Chr16, *Ccdc80* (F: TCGCAGAATGCCAAGGAG

TC; R: CAACAAACACGTCAACGCCC), 350 bp; Chr17, *Oct4* (F: ACGGGTGGG TAAGCAAGAAC; R: TCACCGGACACCTCACAAAC), 187 bp; Chr18, *Arhgap12* (F: ACACAATCTCCCCAGGAGTG; R: CACACCCGCTCCTAACATCT), 204 bp; Chr19, *Aip* (F: TTTGAGACAGGGGGCCTTAC; R: CTGTATGCACGGTGATG GTG), 293 bp; ChrX, *Ccdc22* (F: AGAGCCCTCACAACCAGCTA; R: CCAGATG TTGTGGGTGAGTG), 209 bp

Primers for human: Chr1, *AHDC1* (F: ATTCCGGCTTCAACTCTGGG; R: AG GCCAAAGATCCTGTGTCG), 199 bp; Chr2, *HDAC4* (F: TTCCTGTCTCCTCCA AAGGCAG; R: TGCAGCACATGGTCTTACTGG), 148 bp; Chr3, *LZTFL1* (F: AC AACAAGCCCTATGCTCCA; R: GGCCTGGTTTTCTCAGACGA), 299 bp; Chr4, *SPOCK3* (F: TCTGCATGAGACTGCCATCC; R: CAGCTAGTGCTTGGGATC GT), 384 bp; Chr5, *CENPK* (F: TGGCTTTTGGAAACTAGCACA; R: CCCCCTA GATGAGTATCTGGC), 328 bp; Chr6, *ZFP57* (F: TTGGGGCTCCGAAACAAC TT; R: CCTCATTCACTGTAGAAGGCAAG), 310 bp; Chr7, *TFR2* (F: GGTGGCA AGATGGGGATTCT; R: CTGCACAGCAACAACTGTCC), 358 bp; Chr8, *CNOT7* (F: TGACCCAAGTAAGTGTGGAGC; R: GTGGACTCAAGCCATTCCTCT), 350 bp; Chr9, *WNK2* (F: ACCATTGTGCCAAATGCACC; R: AGGGGTGGCCC TCATATTCT), 195 bp; Chr10, *VCL* (F: TCTGCCGGTGTGTTAACCTG; R: TGG GTGGACAACTAGCAAACA), 309 bp; Chr11, *NCR3LG1* (F: CTGTGAGGGTAT GGTAGAAGCC; R: AGTTGCTCCTCAAGGGGGTA), 399 bp; Chr12, *POLE* (F: AGCTTCCGAGACATGGAACG; R: TGAAATTGCAGCCTTCTGCG), 190 bp; Chr13, *ENOX1* (F: TTACCAGGCACCTGTCACCT; R: AACAAACACCCTTG CCTATGGA), 260 bp; Chr14, *HIF1A* (F: AAGGTGTGGCCATTGTAAAAACTC; R: CATCAGTGGTGGCAGTGGTA), 252 bp; Chr15, *SPATA8* (F: TCAATGTGGG CTCCTGATGC; R: CACCCATCCAGGTTCAGGAAA), 260 bp; Chr16, *TOX3* (F: CACGCCATCTTGTTCCACGA; R: TGCAATGTCTTTCTCTTTCCCTCC), 314 bp; Chr17, *ELAC2* (F: GAGCCCACCAACTACCAACA; R: TGGGCGGGC TTATTTGGTTT), 304 bp; Chr18, *RAB12* (F: TGGTGCTCACTTTGGGCATT; R: GCAGAGAGTTCACATTGGACAG), 231 bp; Chr19, *EPOR* (F: TTACCCTT GTGGGTGGTGAAG; R: GTACTCCTCTGCCTCCATTGT), 210 bp; Chr20, *DHX35* (F: AACTCCGAGCCTACAATCCC; R: CCTTGTGAGGAAGGCCTGT AT), 253 bp; Chr21, *APP* (F: TTGGGCGAGGTCTTGTAGAAT; R: GAGGTGGT TCGAGGTAATCCA), 278 bp; Chr22, *MORC2* (F: TTGACGAGAGTGTTGGCA GG; R: TCAGTGTTCTCACTCTCTGTGG), 270 bp

**Genotyping of single cell.** The *Brca1^{co/co}* and *Brca1^{△exon11/△exon11}* allele was confirmed by PCR amplification with primers F1/R1 (F1: CTGGGTAGTTTGTA AGCATCC, R1: CAATAAACTGCTGGTCTCAGG), F1/R2 (F1: CTGGGTAGTT TGTAAGCATCC, R2: CTGCGAGCAGTCTTCAGAAAG) using whole genome-amplified single-cell DNA, yielding products of 470 bp and 621 bp for the wild-type and the *loxP*-flanked alleles, respectively. Cells with the genotype of *Brca1* for wild type and *Brca1^{△exon11/△exon11}* were used for further experiments. The PCR reaction was performed using MyTaq Red Mix (Bioline, #BIO-25044) following the thermocycling conditions: one cycle of enzyme activation at 95 °C for 5 min, 35 cycles of denaturation at 95 °C for 30 s, and annealing at 60 °C for 30 s, extension at 72 °C for 1 min, 1 cycle of final extension at 72 °C for 5 min, following hold at 4 °C.

**NEB library preparation.** Before library construction, 100 ng (single cell) or 1 μg (bulk sample) of DNA was acoustically sonicated to 200–500 bp using the Covaris Sonicator S220. A MinElute PCR Purification Kit (Qiagen, #28006) was used for the purification of fragmented DNA. Libraries were constructed using the NEB-Next Ultra DNA Library Prep Kit for Illumina (NEB, #E7645) following the manufacturer's protocol. AMPure XP (BECKMAN COULTER, #A63881) was used for the purification of libraries. The size and concentration of libraries were measured by a 2100 Bioanalyzer (Agilent). The final concentration was confirmed by quantitative real-time PCR using a KAPA Library Quantification Kit (KAPA Biosystems, KK4835).

**Exome capture.** Exome capture of samples from mice was performed for libraries using the SeqCap EZ Exome Library SR Platform (Roche NimbleGen), including the SeqCap EZ Developer Reagent (Roche, #06471684001) and the SeqCap EZ Reagent Kit Plus v2 (Roche, #06977952001), following the manufacturer's protocol. The capture platform targeted an over 54.3 Mb of region, including the exons, promoters, and UTRs. Exome capture of samples from humans was performed for libraries using the TruSeq Rapid Exome Kit (Illumina, #FC-144-1004) following the manufacturer's protocol. The capture platform targeted an over 64 Mb of region, including the exons, promoters, and UTRs.

**Next-generation sequencing.** The whole-exome libraries were sequenced using an Illumina HiSeq 4000 or X Ten platform. Data were processed by using the CASAVA 1.8.1 pipeline (Illumina Inc.), and sequence reads were converted to FASTQ files for downstream analysis.

**Sequence alignment and processing.** The alignment of the sequenced reads to the mouse genome (GRCm38.84) or the human genome (GRCh38.p12) was done by employing the BWA-MEM algorithm from the Burrows–Wheeler Aligner software package version 0.7.12-r1039[68]. The duplicate reads were marked using PICARD (http://broadinstitute.github.io/picard/) version 2.5.0. Realignment around indels using GATK IndelRealigner was implemented to reduce mismatches

and improve the read alignment. The base quality scores were recalibrated to correct the over- or under-estimated scores using GATK BaseRecalibrator. The above tools of GATK[69] were executed in GATK version 3.6.

**SNVs detection in single cells and bulk samples.** SNVs for bulk samples were called using the MuTect2 program[70]. The variants were filtered with 20 minimum coverage, 5 minimum variant-supporting reads, and the recommended arguments provided by GATK. Single cells and their paired bulk tissue were put through somatic variant calling using SAMtools[71] version 1.3 followed by VarScan[72] version 2.3.9. Somatic mutations were classified as high confidence using processSomatic with 10% maximum variant allele frequency in the normal sample. An SNV was filtered by somaticFilter with 10% minimum variant allele frequency and *P* value < 5e−3. Also, for the bulk samples 20 minimum coverage and 5 minimum variant-supporting reads whereas for single cells 10 minimum coverage, 3 minimum variant-supporting reads, and presence in at least three single cells were set. To further reduce the possibility of germline mutation calling, out of all the SNVs found in single cells, the SNVs which were unique to one mouse were retained for further analysis. To ensure the identified SNVs above were captured in all cells analyzed, the identified mutation sites were checked for all single cells, the SNVs with the same mutation, and 2 minimum variant-supporting reads were retained. The data from variant calling were summarized and plotted into oncoplots using R package ComplexHeatmap 2.0.0[73].

**Copy-number detection in bulk samples.** To estimate the somatic CNVs of bulk tissue samples, CNVkit version 0.8.5[50] was used on aligned paired tumor-normal WES data. The CNVs were inferred applying the standard procedure with default parameters (https://cnvkit.readthedocs.io/en/stable/pipeline.html).

**Copy-number detection in single cells.** The relative coverage of many genomically adjacent genes would average out gene-specific coverage pattern and provide profiles which illuminate CNVs[49]. Combined exonic RPKM values were calculated using samtools version 1.3 and bedtools version 2.19.0 for all genes. The normalized values were sorted firstly by their genomic locations, starting from chromosome 1 to chromosome 19 (for mouse) or chromosome 22 (for human) and ending with chromosome X and then by the gene start position. Using a moving average of 100 analyzed genes chromosomal CNVs in each cell and at each analyzed gene were estimated[49]. All the values were centrally scaled before plotting. The CNVs with values ranging from −1 to 1 were removed to deemphasize low-amplitude CNVs in cells from mice. For mouse samples, the WT mammary gland bulk was used as a control, the regions that were also covered in WT mammary gland bulk sample with a round-off value of more than 1 or less than −1 were removed from the analysis. The majority of tumor cells have amplification in chromosome 11 in mouse 476. To define large amplification of chromosome 11 (driver events) in each cell (Fig. 5b), a simple calculation was performed as follows: the mean(a) of CNV of chromosome 11 was calculated by averaging the scaled RPKM value of genes on chromosome 11 for each cell, the mean(t) of CNVs of tumor cells was calculated by averaging the mean(a) for all tumor cells. When mean(a) > mean(t), it was counted as amplification, otherwise not. To show the major difference of CNVs between TM00089 and TM00091 tumor, the CNVs with values less than −1.5 or more than 1.5 were maintained.

**Mutational signature analysis.** Using a simple data frame containing genomic position, base change for each mutation and sample identifier mutational signatures were generated using the deconstructSigs[74] package 1.8.0. The hard coded Bioconductor library BS.genome.Hsapiens.UCSC.hg19 was changed to BSgenome. Mmusculus.UCSC.mm10::Mmusculus to determine statistically the contribution of each signature in each mouse tumor.

**Phylogenetic analysis.** Phylogenetic analysis inferred from multi-site mutational profiles provided an insight into the intratumor heterogeneity which can be observed among the single-cell populations in different mice. All the nonsynonymous mutations were converted to a binary present/absent matrix for each single cell sample and used as input to generate an unrooted tree using parsimony ratchet method implemented in the R package phangorn 2.5.5[75]. The length of the branches is determined using the function acctran and is proportional to the number of nonsynonymous mutations. Clonal lineage was analyzed with Time-Scape[76] based on several potential driver genes and their clonal frequencies across the developmental time points in the mouse.

**Gene Ontology term and pathway association.** The genes with SNVs plotted in oncoplots were annotated with GO term, Reactome, or KEGG pathway. The pathways enrichment of the genes with CNVs were analyzed using DAVID–KEGG Bioinformatics Resource[51].

**Target amplification and Sanger sequencing.** To validate the SNV calling from bulk tissue and single cells, we designed primers and performed PCR to amplify SNV target sites from the original DNA material followed by Sanger sequencing.

Primers for some critical SNVs present in both bulk and single cells were designed (Supplementary Data 4). Unipro UGENE 1.30.0 was used for sequence analysis.

**Droplet digital PCR**. Digital PCR was performed on a QX200 ddPCR system (BioRad). To examine copy number amplification in the mammary gland and tumor tissue, 45 ng of genomic DNA was used for each reaction. PCRs were prepared with ddPCR Supermix for Probes (BioRad, #1863026) at a final concentration of 900 nM for the primers and 250 nM for each probe and partitioned into a median of 20,000 droplets per sample via a QX200 droplet generator (BioRad, #1864002) following the manufacturer's protocol. The primers and probes were summarized in Supplementary Data 6. All genomic DNA was digested directly in the ddPCR reaction with the BamHI restriction enzyme (New England BioLabs, #R0136S) following the manufacturer's protocol. Three negative controls with no DNA template were used for each batch. PCRs were performed on a thermal cycler with a 96-deep-well reaction module (BioRad, C1000 Touch) following the specific cycling conditions: one cycle of enzyme activation at 95 °C for 10 min with a 2 °C/s of ramp-up rate, 40 cycles of denaturation at 94 °C for 30 s, and annealing/extension at 60 °C for 1 min with a 2 °C/s of ramp-up rate, and 1 cycle of enzyme deactivation at 98 °C for 10 min with a 2 °C/s of ramp-up rate, followed by a hold at 4 °C with a 1 °C/s of ramp-down rate. Plates were read on a QX200 droplet reader (BioRad, #1864003) using QuantaSoft software (BioRad) following the instruction manual. Analysis of the ddPCR data was also conducted using QuantaSoft software 1.7.4 (BioRad).

**Viral production and transduction of cells**. For targeting of murine *Arhgef11* gene, the oligos (F1 5′CACCGTCACCCCCAAAATGGGCCGC3′ and R1-5′ AAA CGCGGCCCATTTTGGGGGTGAC 3′, F2-5′CACCGCACTCACCTGCGGCCC ATTT3′ and R2-5′AAACAAATGGGCCGCAGGTGAGTGC3′) were cloned into the lenti-CRISPR/Cas9v2 vector (Addgene, #52961) following the Zhang lab protocol[77]. For targeting of murine and hominine *Plekha5/PLEKHA5* gene, the oligos (F-5′CACCGCAGAGTTCTCATTAGACCCG3′ and R-5′AAACCGGGTCTAAT GAGAACTCTGC3′ for mouse, F-5′CACCGTCCGGTGACCACCGGCCTCGC3′ and R-5′AAACGCGAGGCGGTGGTCACCGGAC3′ for human) were used. The lentiviral plasmid, envelope plasmid (pMD2.G), and packaging plasmid (psPAX2) were transfected together into 293FT cells with PEI to produce viruses. The culture medium was collected and filtered with a 0.45-μm filter at 72 h after transfection. The viral media was 100× concentrated via PEG precipitation, resuspended with PBS, and saved for infection. Target cells were infected with virus together with 8 μg/ml polybrene. Positive cells were selected for 7 days with puromycin 72 h after infection. All cells used for metastasis experiments were stably labeled with a GFP-expressing vector.

**Arhgef11 and Plekha5/PLEKHA5 sequence analysis**. For validation of target modification, genomic DNA was isolated from cell lines and tumor tissues. Following PCR amplification of murine *Arhgef11* (F1: CGTAGCGTCCAGTGACTA CA and R1: TTGATGACCTCTTGCCGGTC), murine *Plekha5* (F: CTGTTCCT TTGTTGCCTGCC and R: TGCCCGTCCTTCTGAAATCC) and hominine *PLE-KHA5* (F: GTGTCTGCCCCTTCTCTCAC and R: CCGTCTCCAAGTGCTGAT GA), PCR products or TA-cloning products using PCR products were sequenced. Allele modifications were determined by using the control cell as a reference sequence. Unipro UGENE 1.30.0 was used for sequence analysis.

**Migration and invasion assays**. Transwell migration and invasion assays were conducted using the Corning BioCoat Matrigel Invasion Chamber (# 354480) and Control Insert (#354578) according to the manufacturer's instruction. Briefly, $1.5 \times 10^5$ (for B477) cells or $7.5 \times 10^4$ (for MDA-MB-231) cells in DMEM medium without FBS were added to the upper chamber and allowed to migrate (migration assay) or invade collagen-coated membranes (invasion assay) for 22 h at 37 °C, 5% $CO_2$ atmosphere toward DMEM medium with 10% FBS (Prime) (ExCell Bio, #FSP500), 5 μg/ml insulin (Sigma-Aldrich, #I1882), 500 ng/ml hydrocortisone (Sigma-Aldrich, #H0888), 10 ng/ml EGF(Invitrogen, Thermo Fisher Scientific, #13247-051). The cells in the upper chamber were removed with cotton swabs. The cells on the lower surface of the membrane were fixed in 4% paraformaldehyde and stained with crystal violet for photographs under a microscope and quantification using image J 1.51.

**Cell implantation experiment**. In tumor growth experiments, 6–8-week-old BALB/c female mice were used, $4 \times 10^6$ 4T1 cells per mouse were suspended in 60 μl of PBS for mammary fat pad injection. In metastasis experiments, 6–8-week-old nude female mice were used. For B477-GFP cells, G600-GFP cells, and MDA-MB-231-GFP cells, $5 \times 10^5$ cells per mouse, respectively, were suspended in 60 μl of PBS for mammary fat pad injection. The tumor volume was measured every 3 days. At the end of the experiment, lungs, livers, and brains were dissected and fixed in a 10% formalin solution, and the GFP-positive spots were counted using Image J 1.51.

**IHC staining**. Staining of Kras (Abcam, #ab180772), Nras (Abcam, #ab206969), Tpr (Abcam, #ab84516), Akt3 (Abcam, #ab152157), Vimentin (Abcam, #ab92547),

Hepatocyte (HepPar1) (Dako, #M7158), Ppm1d (Abcam, #ab31270), Cxcr4 (Abcam, #ab124824), Arhgef11 (Santa Cruz Biotechnology, #sc-166740), and Plekha5 (Santa Cruz Biotechnology, #sc-390311) were performed on formalin-fixed paraffin-embedded (FFPE) tissue using Histostain-Plus Bulk Kit (Invitrogen, #85-8943) according to the manufacturer's instruction. Briefly, Paraffin-embedded mammary gland, PT, liver, or lung samples were sliced into 4 μm thickness. Next, slides were treated with xylene and followed by 100% alcohol treatment for deparaffination. After treated with peroxidase quenching solution (3% hydrogen peroxidase in methanol) for 10 min to block endogenous peroxidase, the slides were boiled in epitope retrieval buffer at 96–100 °C for 20 min for antigen retrieval. Following washes with PBS and incubation with blocking solution (Reagent A) for 10 min, slides were then incubated with primary antibody (1:100–500 dilution) for 1 h. After washed with PBS, slides were further incubated with the biotinylated second antibody (Reagent B) for 10–20 min and followed by enzyme conjugate (Reagent C) treatment for 10 min. Sections were stained with DAB and then counterstained with hematoxylin.

**Immunofluorescence staining**. The staining of CK14 (Abcam, #ab49806) was performed on cryosection of liver tissues with metastasis from block embedded in optimal cutting temperature compound (OCT). The sections were fixed for 5 min with 4% paraformaldehyde at room temperature and then washed with PBS and PBST (0.25% TritonX-100/PBS) 5 min for each. After blocking with animal-free blocker (Vector Laboratories, #SP-5030) for 1 h, sections were incubated with primary antibody (1:100 dilution) at 4 °C overnight with a parafilm in a humid chamber. Following washes with PBST, sections were incubated with goat anti-mouse IgG secondary antibody (Invitrogen, A32727) (1:1000) for 2 h and counterstained with 4′,6-diamidino-2-phenylindole (DAPI) (Thermo Fisher Scientific, #D1306) for 5 min at room temperature in a dark humid chamber.

**Hematoxylin and eosin (H&E) staining**. For the H&E staining of the tissue section, standard protocol[78] was used with some modification. Briefly, FFPE tissue sections were deparaffinized and rehydrated with xylene for 10 s and 100, 95, 85, 70, 50, 30% of ethanol and water for 1 min each. After stained with hematoxylin (Leica Biosystems, #3801560) for 30 s, slides were washed with water for 30 s, destained with acid ethanol for 1–5 s, then treated with Scott's water. Following staining with eosin Y (Thermo Fisher Scientific, #6766009) for 2 min, slides were dehydrated with 70% alcohol for 10 s, 90% ethanol for 10 s, two change of 100% alcohol for 2 min each. Before mounting with mountant (Sigma, #06522-100 ML), slides were treated with two changes of xylene to extract alcohol.

**Reanalysis of data from breast cancer patient cohort**. Microarray gene expression data including clinical outcomes of 155 breast cancer patients bearing PTs with or without metastasis were retrieved from Gene Expression Omnibus (GEO) under accession number GSE9893[57]. The normal expression of *PLEKHA5* was defined as $\log_2$ expression value $\geq 0$ and the low expression of *PLEKHA5* was defined as $\log_2$ expression value $\leq -0.5$. Kaplan–Meier curves were generated using GraphPad Prism 6.0.

**Statistics and reproducibility**. All representative experiments (such as micrographs, transwell assay, ddPCR, etc.) were performed in triplicates independently. The results were reported as the mean ± SD, statistical significance was calculated using a two-tailed Student's *t* test, unless otherwise indicated. For mouse tumor-free survival and human overall survival analysis, statistical significance was calculated using a two-tailed log-rank test (Mantel–Cox). *P* values were considered statistically significant if the *P* value was <0.05. For all figures, ****indicates *P* value < 0.0001. Error bars represent SD.

**Reporting summary**. Further information on research design is available in the Nature Research Reporting Summary linked to this article.

## Data availability
The NGS raw data that support the findings of this study are available in the Sequence Read Archive, National Center for Biotechnology Information, under accession number SRP279585 (Bulk WES of *BRCA1*-associated mouse models), SRP278027 (Single-cell WES of *BRCA1*-associated mouse models), SRP277871 (Bulk WES of *BRCA1*-associated human PDX models), SRP278032 (Single-cell WES of *BRCA1*-associated human PDX models). The information of samples and the NGS data is provided in Supplementary Data 1. Somatic mutations are provided in Supplementary Data 2, 5. CNVs are provided in Supplementary Data 3. Sanger sequencing results and primers for SNVs are provided in Supplementary Data 4. Primers and probes for ddPCR are provided in Supplementary Data 6. The SNV and CNV data of the patients with *BRCA1* germline mutations of WSI dataset were downloaded from the ICGC (https://dcc.icgc.org/) based on the sample ID and the raw data can be accessed under the accession number EGAS00001001178[21]. Microarray gene expression data including clinical outcomes of 155 breast cancer patients bearing PTs with or without metastasis were retrieved from GEO under accession number GSE9893[57]. The source data underlying Figs. 1a, b, 2a, b, 3e, f, 4a, c–e, 5e–i, 6c, e, g, i, k, l and Supplementary Figs. 1a–f, 3c, e–h, 5j–m, 6g–l are provided as

Source data file. All other data and materials can be requested from the corresponding author upon reasonable request. Source data are provided with this paper.

## Code availability

The code used in the study is available at: https://github.com/Radhav-XuLab/JianlinLiuSingleCell.

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

## Acknowledgements

We thank members of Xu laboratory for helpful advice and discussion; the Animal Research Core for providing the animal housing; the Genomics, Bioinformatics & Single Cell Analysis Core for helping on genomics analysis, and the Information and Communication Technology Office for providing the HPC for data processing. This work was supported by a multiyear research grant (MYRG2016-00138-FHS, MYRG2017-0008-FHS, and MYRG2019-0064-FHS) by the University of Macau, Macau SAR, China; The Science and Technology Development Fund (FDCT) grants (027/2015/A1, 029/2017/A1, 0101/2018/A3, and 0011/2019/AKP) of Macau SAR, China.

## Author contributions

J.L.L., C.X.D., and X.L.X. designed the experiments; C.X.D. and X.L.X. supervised the work; J.L.L., K.M., L.H.M., H.S., Xin.Z., and K.T. performed experiments for bulk tissue and single-cell sequencing; J.L.L. conducted SNV and CNV validation experiments using PCR-Sanger-sequencing or ddPCR; J.L.L., S.M.S., U.I.C. Xin.Z., J.X., X.D.S., and J.J.L. conducted implantation experiments using cell lines and related data collections; J.L.L., S.M.S., and U.I.C. conducted H&E staining, IHC staining, and IF staining; J.L.L. and R.A. carried out all analyses of publicly available data and bioinformatic analysis, L.P. and X.Y.L. also contributed to this part; J.M.Z. contributed scripts for single-cell CNVs analysis and Xu.Z. contributed scripts for clonal lineage analysis with TimeScape. J.L.L. and R.A. wrote the paper, K.M., J.M.Z., H.S., and K.H.W. reviewed the paper, J.L.L., C.X.D., and X.L.X. revised the paper.

## Competing interests

The authors declare that they have no competing interests.
