## [Peer Review File · Nature Communications]

Reviewers' comments:

Reviewer #1 (Remarks to the Author):

Comments for Author

Characterisation of Plekha5/PLEKHA5 as a metastatic suppressor in breast cancer is an important research work done by Liu et al. Interestingly, Plekha5 plays this role regardless of the Brca1 status. Using public data base, authors found out that the expression level of PLEKHA5 was significantly lower in the primary tumours with metastasis compared with the primary tumours without metastasis. Further analysis revealed that the low expression of PLEKHA5 was also associated with worse overall survival in these cohorts of patients.

This study is also a technical advancement in the field of single cell genomics. Authors were able to use ScWES data for simultaneous SNV and CNV analysis. Unfortunately, this approach was under-utilised by authors as the final number of cells interrogated per sample are too small for studying heterogeneity in detail.

I request authors to address following weak points in the manuscript:

1. Authors claim that they have "discovered an evolution process through which tumours initiate from cells with SNVs affecting driver genes in the premalignant stage and then malignantly progress via CNVs acquired".

o Firstly, it is important that authors clearly define which mouse samples are premalignant samples. The sample labels are not explained properly in Figure legends. On page 24, authors describe VMMG as early stage and 476MMG as late stage of malignant transformation. Are these both premalignant or only the early stage sample is considered as premalignant? Also, are VMMG mice littermate controls for mouse tumours #153, # 476 and #576?

o Was SNV and CNV analysis done between wildtype-mammary gland and premalignant mammary gland (samples corresponding to Figure 3a)? Is CNV profile of wildtype mammary gland different from that of premalignant cells?

o Although analysis between premalignant and malignant stages in mouse model was done, but a similar study utilising BRCA1 deficient pre-neoplastic normal tissue and BRCA1 tumour is required to understand the role of SNVs and CNVs during human breast cancer progression.

o Overall, small number of cells were analysed for Single cell SNV and CNV analysis in this study. To strength this body of work, a throughput single cell Exome analysis approach is recommended.

2. Is there any role played by PLEKHA5 in normal mammary gland development since its expression level was found to be significantly higher in normal tissue compared to breast tumours studied from the TCGA database?

3. Regarding BRCA1 WT TNBC PDX and BRCA1 TNBC PDX – Were they both generated from primary breast tumours or metastatic lesions? If one was from primary and other from metastatic, will it affect the conclusions drawn from these datasets including bulk cell and single cell profiles?

4. It is proposed that chromosomal changes (CNVs) drives premalignant cells towards malignancy. Is this just a correlation or cause of malignancy?

5. Figure 1:

o Explain the tumour sample labels in figure legend. e.g what does – 5L, -5R, etc mean?

6. Figure 3:

- o Is the picture shown in Figure 3a (first panel) representative of 4wk gland? It looks more like an adult gland since mammary ductal tree seems to fill the entire fat pad.
- o What criterion was used to define premalignant mammary gland (Fig 3a; second panel)
- o The scale bar is not clearly visible in Figure 3a (third panel)
- o While discussing Figure 3 results, authors mention that TNBC PDXs (WT and BRCA1) were included in single cell WES analysis, however these results are not discussed in this section. It is confusing for readers.

7. Supp Figure 3:

- o In panel a, the fluorescent thumb images are quite small. Reducing top bar-plot and increasing the thumb image size will be helpful.
- o Label band size and ladder information in panels d and e.

8. Figure 4:

- o In Figure legend or result section, please provide explanation how it was determined that 476PT contain at least three major sub-clones (Figure 4d) and 153PT is relatively homogenous?

9. Supp Figure 4:

- o In panel d, please label which cells belong to primary tumour and which ones are from LMT.

10. Figure 5:

- o Provide scale bar information for panels h-k in figure legend.

11. Supp Figure 5:

- o Authors discuss that VMMG and 476MMG show higher heterogeneity than 476PT, 153PT and 153LMT (panels, a-e). Please provide evidence to support your claim because there does not seem to be much difference between 476MMG and three tumours (just by looking at the panels). My understanding is that authors will need bigger cell numbers to robustly compare heterogeneity across all the five samples.
- o Mouse cells were captured and processed using the C1 platform but why a different approach was used for human PDX samples?

12. Figure 6:

- o Provide scale bar information for panels a-c, and g in figure legend.
- o Panels k and L – Could this information be broken down based on cancer subtype of the patients. Similarly, for Sup Fig 6h – j.

13. Overall, the figure legends lack crucial details about experimental results shown in the corresponding figures.

Reviewer #2 (Remarks to the Author):

This study uses single-cell whole-exome sequencing to address the question of etiology in BRCA mutant cancers. The experiments described were carefully performed and the authors ensure good DNA quality and that multiple criteria were implemented to minimize false positive. The combinatorial analyses of single nucleotide variation (SNV) and copy number variation (CNV) in single cells from different tumor progression stages in Brca1-deficient mice revealed an interesting association of increased CNV, specifically amplification or deletion patterns, in malignant tumors. Using this

technique, the authors identified Plekha5 as a tumor metastasis suppressor based on their experiments that demonstrated that Plekha5 depletion enhanced cell migration and invasion. As noted by the authors, their results are different from previous studies that showed Plekha5 knockdown prevents blood-brain barrier transmigration and invasion; however, the functional assays investigating the role of Plekha5 presented in this work in figure 6 are convincing.

Major comments:

1) Bulk whole-exome sequencing of 23 breast tumors from Brca1-deficient mice in figure 1 and supplementary figure 1 identified C>A/G>T transversion mutation as the major mutation type. However, in single-cell whole-exome sequencing data from different tumor developmental stages in Brca1-deficient mice (total n=135), C>A/G>T signature appears to constitute a relatively minor fraction (Supplementary figure 3h). Instead, C>T/G>A transition mutation is the predominant change in single-cell sequencing data. It is not clear how the authors reconcile this discrepancy. Is this mutation signature difference due to sample size or tumor stages or others?

Overall, this is an interesting study that could be acceptable for Nat Comm. I have only a few comments.

Minor comments:

1) In the introduction section of the manuscript on page 4, the authors described that "Single-cell whole-genome sequencing has been used for copy number variation (CNV) analysis in several studies". The authors should cite seminal work from this field. The same is true for this statement in the same paragraph: "Whole-exome sequencing was previously used to obtain high coverage for confident SNV calling in single-cell analysis".

2) On page 5, it should be Wellcome Trust Sanger Institute instead of Welcome Trust Sanger Institute.

Reviewer #3 (Remarks to the Author):

The authors are using both bulk and single cell WES of BrCa1 deficient mammary glands and tumors to identify SNV and CNV that might be associated with tumor initiation. The manuscript is well written and moderately easy to follow, but is dense with data that makes reading the paper a long process. Overall, this is a detailed and comprehensive analysis of subsequent mutations that develop in BrCa1 mutant tumors that advances our knowledge of the tumor-initiating events subsequent to having a germline mutation in BrCa1.

Comments:

1. Introduction, page 3: In the first paragraph there is a sentence stating that drug resistance can be a problem for PARPi effectiveness. This sentence should be prefaced by a new sentence stating that PARP inhibitors have been found to be very effective in control of BRCA1 mutated tumors, however, drug resistance can still develop and reduce effectiveness of this therapy.

2. In general, many figures are so detailed and small that they are almost impossible to decipher, even when enlarged. For example, Figures 1c and d, Supp Figures 2b and 3f.

3. The H&E images in Figure 2c do show different morphologies but the arrows pointing to different features are not convincing. To convince the reader about the large number of neutrophils, the authors need to stain with a neutrophil specific marker. Aggressive mesenchymal tumor cells are not

obvious to me in the second image. The third arrow points to an "amorphous tumor cell". What is that? Finally, I'm not convinced by what the fourth and fifth arrows are showing. In summary, I do not find that the conclusions made from these H&E images are convincing.

4. Page 13 and Figure 2d. The authors show IHC images for the presence of proteins identified as carrying mutations. The antibodies are positive for the proteins, but are these antibodies specific for the mutant form of the protein? If not, then the detection of the protein does not "suggest that these mutations may indeed act as driving forces for the progression of these tumors". The wild type protein is likely to be detected as well.

5. Page 13: How similar was 153RT to 153PT? The statement is made that these two tumors have different driver genes but I can't find the data for 153RT. Does this tumor carry the Plekha5 mutation or the Arhgef11 mutation?

6. Supp Figure 3c: please mark the lanes. The explanation of each lane is not clear to me.

7. Y axis of figure 4A should be labelled "Number of nSNVs"

8. Following bulk sequencing, the authors have moved on to single cell sequencing, and imply in the text on page 14 that they have generated another mouse model of BrCa1 defective tumors, this time with fluorescence markers to enable BrCa1 WT and BrCa mutant cells to be detected within the same tumor. However, when labelling the resulting tumors, the numbers are the same as those used for the bulk sequencing, so I assume that even for the bulk sequencing, the authors used the mT/mG mice? This is not made clear in the text.

9. I'm not convinced that Figure 3e is necessary.

10. Page 24: the sentence beginning "Gbp4 was reported to be involved in the innate immune system..." needs to be re-written as it indicates that IFN γ negatively regulates IFN α , rather than Gbp4 regulating IFN α .

11. Page 24: the authors go on to suggest that Gbp4 may drive tumor initiation and formation, but where is the evidence that the mutation in Gbp4 has functional consequences – that the mutation increases the activity of the protein?

12. Page 24: It is surprising that a mutation Plekha5 was not detected by bulk sequencing when it was contained in 50% of the 153PT cells assessed at the single cell level. Do the authors have an explanation for this discrepancy?

13. Page 25: The authors note that Plekha5 was detected on 0.5% of BC patients in TCGA. Did this correlate with patients who also carried the BrCa1 mutation?

14. Supplementary Figure 5j-m: What is the significance of the different colors of the bars in these graphs?

15. What was the rationale for choosing a functional study on Plekha5 instead of Arhgef11 or one of the other more prominent somatically mutated genes?

16. Can the authors clarify the difference between the OS curves for BC patients in Figure 6i and SuppFigure 6j. I assume that these data are from different cohorts of patients, but the outcome data for those with "normal" expression of Plekha5 are very different for the two cohorts. In SuppFigure 6j,

there are no survivors in either group after 4000 days. This is unusual for BC, unless this was a cohort with metastatic cancer at diagnosis.

17. Can the authors speculate on how Plekha5 suppresses metastasis? They have shown increased migration and invasion, but is anything known about pathways in which Plekha5 functions, especially in light of the recent paper from Hongyi Zhang et al (2020) Cancer 126(5):1016-1030?

18. There are other studies analysing the genomic changes subsequent to BrCa1 loss in breast cancer, but probably not in the detail described here. However, these previous studies should be included in the Discussion, especially the papers from J. Jonkers and colleagues.

Reviewers' comments:

Reviewer #1 (Remarks to the Author):

Comments for Author

Characterisation of Plekha5/PLEKHA5 as a metastatic suppressor in breast cancer is an important research work done by Liu et al. Interestingly, Plekha5 plays this role regardless of the Brca1 status. Using public data base, authors found out that the expression level of PLEKHA5 was significantly lower in the primary tumours with metastasis compared with the primary tumours without metastasis. Further analysis revealed that the low expression of PLEKHA5 was also associated with worse overall survival in these cohorts of patients.

This study is also a technical advancement in the field of single cell genomics. Authors were able to use ScWES data for simultaneous SNV and CNV analysis. Unfortunately, this approach was under-utilised by authors as the final number of cells interrogated per sample are too small for studying heterogeneity in detail.

We would like to thank Reviewer 1 for the appreciation of our work and for pointing out the important technical advances of the work.

I request authors to address following weak points in the manuscript:

1. Authors claim that they have “discovered an evolution process through which tumours initiate from cells with SNVs affecting driver genes in the premalignant stage and then malignantly progress via CNVs acquired”.
 - o Firstly, it is important that authors clearly define which mouse samples are premalignant samples. The sample labels are not explained properly in Figure legends. On page 24, authors describe VMMG as early stage and 476MMG as late stage of malignant transformation. Are these both premalignant or only the early stage sample is

considered as premalignant? Also, are VMMG mice littermate controls for mouse tumours #153, # 476 and #576?

We agree with Reviewer 1 that it is very important to clearly define the premalignant samples. We are sorry about the confusions caused. In our study, we define the mammary gland samples in *Brca1* deficient mice before tumor formation as the premalignant samples, more specifically, we define the *Brca1*-mutant cells isolated from mammary glands with *Brca1* deficiency as the premalignant cells. Here, VMMG means 'virgin mutant mammary gland cell' and 476MMG means '476 mutant mammary gland cell', so they both are premalignant samples. In addition, VMMG is from a 4-month old virgin mouse bearing no tumor and 476MMG is from an 11-month old mouse bearing tumor, so VMMG is defined as 'early stage' and 476MMG is defined as 'late stage'. And here 'stage of premalignant transformation' may be more precise than 'stage of malignant transformation'. The detailed explanations for the sample labels have been added into Figure legend of Figure 3e and the change of 'stage of premalignant transformation' has been made in the main text of the revised manuscript (Page 24, line 428).

VMMG is a group of cells showing the information of the early stage of premalignant transformation upon deficiency of *Brca1*. They are not littermate control for 476 and 153, but they were very closely related as our mouse colony was initially derived from a couple of mating pairs.

o Was SNV and CNV analysis done between wildtype-mammary gland and premalignant mammary gland (samples corresponding to Figure 3a)? Is CNV profile of wildtype mammary gland different from that of premalignant cells?

Yes, the SNV and CNV analysis were done between the wildtype mammary gland and the premalignant mammary gland for samples corresponding to Figure 3a.

Because our goal is to identify somatic SNVs and CNVs caused by *Brca1* mutation, it is important to filter out the background mutations and germline mutations. For bulk DNA

analysis, the SNVs and CNVs were called out from *Brca1* mutant samples by filtering out SNVs and CNVs in wildtype mammary gland and then in the matched kidney for germline mutations filtering because Cre is not expressed in the kidney. For single-cell DNA-sequencing, we were able to identify wildtype cells and *Brca1* mutant cells in the same mammary gland because MMTV-Cre only deletes *Brca1* in about 60% of mammary epithelial cells, and we used same filtering processes as bulk DNA analysis. Our results showed no obvious difference in CNVs between wildtype mammary gland cells and mutant (pre-malignant) mammary gland cells. These data have been included in Figure 5a of the revised manuscript.

o Although analysis between pre-malignant and malignant stages in mouse model was done, but a similar study utilising BRCA1 deficient pre-neoplastic normal tissue and BRCA1 tumour is required to understand the role of SNVs and CNVs during human breast cancer progression.

Yes, it was done. When we mention pre-malignancy, it also includes the mammary glands that have not developed tumors in the tumor-bearing mice. We classified this type of mammary gland might as “*BRCA1* deficient pre-neoplastic normal tissue”. One pair of samples we already used in our comparison, 476 MMG and 476 PT, belong to this type. We have made the comparison and the data was shown in Figure 5a, 5b.

o Overall, small number of cells were analysed for Single cell SNV and CNV analysis in this study. To strength this body of work, a throughput single cell Exome analysis approach is recommended.

The recommendation to use a throughput single cell Exome analysis certainly will provide more information. However, due to some special situations, we decided to use the current approach for this study. First, because the MMTV-Cre gives incomplete deletion, every *Brca1* conditional knockout mammary gland always contains a mixture of wildtype and mutant epithelial cells. Four years ago, when we started this project, we decided to overcome this difficulty by careful genotyping single cells one by one before

using them for sequencing, many *Brca1*-heterozygous cells were excluded for sequencing (Supplementary Figure 3c; Page 16, line 305). Second, because faithfully amplifying genomic DNA for all chromosomes of single cells has been very challenging, we need to quantitatively examine all chromosomes by PCR for all cells after DNA amplification. In this study, from a total of 934 single cells captured, we manually examined all these cells one by one, and only found 135 cells could be used. These features prevented the sequencing of many cells. Now if we would like to use another high throughput approach to sequence more cells, the mixture of cells and representative chromosome coverage will become issues, which would be hard to overcome. Third, in this study, we were also very careful in determining enough number of cells to be used for the analysis by providing a saturation test. Our data indicated that the number of cells used already reached saturation plateau (Fig. 3f). Thus, we feel that most major mutations have already been identified in our assay and adding more cells will cause significant delay and also might not have an obvious impact on mutation spectrum and it also will not change the main conclusion of this study.

2. Is there any role played by *PLEKHA5* in normal mammary gland development since its expression level was found to be significantly higher in normal tissue compared to breast tumours studied from the TCGA database?

We believe you might refer to our analysis of Human Cancer Metastasis Database (Fig. 6k), which revealed that the expression level of *PLEKHA5* was significantly higher in the primary tumors without metastasis compared with the primary tumors with metastasis. The data also revealed that low expression of *PLEKHA5* was associated with worse overall survival in cancer patients (Fig. 6l). We primarily use these human data to support our study in the mice for the role of *PLEKHA5* as a tumor metastasis suppressor. To our knowledge, so far there is no report about the role of *PLEKHA5* in normal mammary gland development. But for studying its role in the mammary gland, it requires the generation of mutant mice with complicated analysis by different approaches. This is a certainly important issue to be further investigated in the future. We have indicated this at the end of the Discussion (Page 42, line 812-814).

3. Regarding BRCA1 WT TNBC PDX and BRCA1 TNBC PDX – Were they both generated from primary breast tumours or metastatic lesions? If one was from primary and other from metastatic, will it affect the conclusions drawn from these datasets including bulk cell and single cell profiles?

PDX-TM00089 and PDX-TM00091 were both obtained from the Jackson Laboratory and they were generated from primary breast tumors. Both are invasive ductal carcinoma, grade 3 TNBC, and actually, both are *BRCA1* mutant PDX. In our old version, we mistakenly indicated TM00089 as *BRCA1* wild type. We are very sorry about this and have revised the sentence as “... from two human *BRCA1*-deficient (TM00089 and TM00091) xenograft breast tumors...” (Page 32, line 572-573). We indicated in the revised Figure legend of Figure 5 that “TM00089, which carries a frameshift variant: *BRCA1*-V757fs; and TM00091, which carries a common pathogenic missense variant C61G in *BRCA1* RING domain.” (Page 31, line 541-542).

TM00089 contains a frameshift variant V757fs:

WT: Human genome (*BRCA1*): ...AGG GTT TTG... (...RVL...)

MT: PDX-TM00089 (*BRCA1*): ...AGG TT TTG... (...RV757-frameshift)

TM00091 contains a common pathogenic missense variant C61G in the *BRCA1*-RING domain:

WT: Human genome (*BRCA1*): ...CAG TGT CCT...(...QCP...)

MT: PDX-TM00091 (*BRCA1*): ...CAG GGT CCT...(...QGP...)

4. It is proposed that chromosomal changes (CNVs) drives premalignant cells towards malignancy. Is this just a correlation or cause of malignancy?

Based on our analyses, CNVs were mainly found in tumor stage and copy-number amplification of well-known oncogenes (*Ppm1d*, *Cxcr4*, etc.) or copy-number deletion of well-known tumor suppressor genes (*E2f2*, *Wnt5a*, etc.) were found in the regions,

which were powerful driving forces for malignancy transformation. So we believe that the loss of *Brca1* initially causes genome instability, which could hit these regions bearing potential tumor suppressors and oncogenes. The amplification/activation of oncogenes and/or inactivation of tumor suppressors could serve as a driving force for tumor formation. Therefore, we believe these CNVs are a cause of malignancy rather than a correlation.

5. Figure 1:

o Explain the tumour sample labels in figure legend. e.g what does – 5L, -5R, etc mean?

We thank the Reviewer for noticing that this information was missing. 5L means 5th left mammary gland and 5R means 5th right mammary gland. This information has been included in Figure legend of Figure 1 in the revised manuscript (Page 8, line 144-147).

6. Figure 3:

o Is the picture shown in Figure 3a (first panel) representative of 4wk gland? It looks more like an adult gland since mammary ductal tree seems to fill the entire fat pad.

The mammary gland in Figure 3a (first panel) is a 4-month gland, so yes, it's an adult gland. This information has been included in Figure legend of Figure 3a in the revised manuscript (Page 15-16, line 269-274).

o What criterion was used to define premalignant mammary gland (Fig 3a; second panel)

Loss of *Brca1* usually causes abnormal morphogenesis of mammary gland ductal tree, and the mutant mammary cells gradually become malignantly transformed and eventually develop into tumors (Xiaoling Xu et al. 1999, Nature Genetics.). So, the criterion we used to define premalignant mammary gland is that the mammary gland in *Brca1*-deficient mouse that has not developed mammary tumor yet.

- o The scale bar is not clearly visible in Figure 3a (third panel)

We have replaced a new image for Figure 3a (third panel) in the revised manuscript.

- o While discussing Figure 3 results, authors mention that TNBC PDXs (WT and BRCA1) were included in single cell WES analysis, however these results are not discussed in this section. It is confusing for readers.

We thank the Reviewer for pointing this out. Because we are not able to get matched normal control of the same patients from PDX models, the reliable somatic SNVs results of human single cells from PDX models cannot be obtained and discussed. To avoid confusing, we adjusted Figure 3a and added one sentence to explain this in the Results of this section in the revised manuscript (Page 19, line 353-355).

7. Supp Figure 3:

- o In panel a, the fluorescent thumb images are quite small. Reducing top bar-plot and increasing the thumb image size will be helpful.

We have adjusted the image for the Supplementary Figure 3a in the revised manuscript.

- o Label band size and ladder information in panels d and e.

We adjusted the order of images in the revised version (reorder d to e, e to f). The information has been added to the Supplementary Figure 3e and f in the revised manuscript. The size for each band can also be found from the Methods, Chromosome PCR panel (Page 45-47, line 881-930).

8. Figure 4:

- o In Figure legend or result section, please provide explanation how it was determined that 476PT contain at least three major sub-clones (Figure 4d) and 153PT is relatively homogenous?

To make it easier to understand for the readers, we replaced two new images for Figure 4d and e. The explanation has been added into Figure legend of Figure 4d and e in the revised manuscript (Page 22, line 388-391).

9. Supp Figure 4:

o In panel d, please label which cells belong to primary tumour and which ones are from LMT.

The labels have been added to Supplementary Figure 4d in the revised manuscript.

10. Figure 5:

o Provide scale bar information for panels h-k in figure legend.

We added some new data in Figure 5, so we moved Figure 5h-k of the old version to Supplementary Figure 5n-q in the revised version. The scale bar information has been provided in the Figure legend of Supplementary Figure 5 in the revised manuscript (Page 28, line 499-501).

11. Supp Figure 5:

o Authors discuss that VMMG and 476MMG show higher heterogeneity than 476PT, 153PT and 153LMT (panels, a-e). Please provide evidence to support your claim because there does not seem to be much difference between 476MMG and three tumours (just by looking at the panels). My understanding is that authors will need bigger cell numbers to robustly compare heterogeneity across all the five samples.

We agree with the Reviewer that bigger cell number will benefit a robust comparison of heterogeneity across all the samples. However, as we replied in point 1, due to multiple reasons, we were not able to add more cells. Instead, we deleted this claim and adjusted the text in the revised manuscript to avoid a strong conclusion drawn by using the data we have (Page 26, line 486-487):

“...a common amplification or deletion pattern of CNVs was only found in the tumor stage (476PT, 153PT, and 153LMT) but not in the premalignant stage (VMMG and 476MMG).”

o Mouse cells were captured and processed using the C1 platform but why a different approach was used for human PDX samples?

The isolation of single cells from human PDX samples was done much later than the isolation of mouse cells. Because we do not have sufficient consumables and reagents for the C1 platform for human samples collection at that time, so we used micro pipetting to collect single cells for PDX model. The difference is at the obtaining single cells and procedure for DNA amplification (WGA), but the principle of WGA is same for C1 and REPLI-g Single Cell Kit (QIAGEN, 150345), they both use MDA method to amplify whole-genome DNA. Once got the single-cell DNA, we used the same procedure for quality control, DNA-sequencing and analysis.

12. Figure 6:

o Provide scale bar information for panels a-c, and g in figure legend.

The scale bar information has been provided in the Figure legend of Figure 6 in the revised manuscript.

o Panels k and L – Could this information be broken down based on cancer subtype of the patients. Similarly, for Sup Fig 6h – j.

We have tried to break down the data based on cancer subtype of the patients. First, we divided the samples into different breast cancer subtypes, then we analyzed the expression of *PLEKHA5* in primary tumors with or without metastasis in different subtypes. Because number of metastasis cases in TCGA dataset is very small, after the breakdown, every subtype group is even smaller for analysis. In GSE9893 cohort 155 tumors have *PLEKHA5* information with metastatic information. Basal group has 14

tumors (no metastasis: 13; metastasis: 1), LumA group has 8 tumors (no metastasis: 5; metastasis: 3), LumB group has 12 tumors (no metastasis: 3; metastasis: 9), Normal group has 10 tumors (no metastasis: 1; metastasis: 9), and Her2 group has 111 tumors (no metastasis: 85; metastasis: 26). Only Her2 group contain enough number of tumors for statistical analysis, so we only analyzed Her2 group for GSE9893 dataset, as shown below (Figure 1 below), which showed the similar result as we showed in Figure 6k and l of the manuscript. Therefore, we still keep the old data in Figure 6k and l.

Figure 1

13. Overall, the figure legends lack crucial details about experimental results shown in the corresponding figures.

We have added more details in the figure legends in the revised manuscript.

Reviewer #2 (Remarks to the Author):

This study uses single-cell whole-exome sequencing to address the question of etiology in BRCA mutant cancers. The experiments described were carefully performed and the authors ensure good DNA quality and that multiple criteria were implemented to minimize false positive. The combinatorial analyses of single nucleotide variation (SNV)

and copy number variation (CNV) in single cells from different tumor progression stages in Brca1-deficient mice revealed an interesting association of increased CNV, specifically amplification or deletion patterns, in malignant tumors. Using this technique, the authors identified Plekha5 as a tumor metastasis suppressor based on their experiments that demonstrated that Plekha5 depletion enhanced cell migration and invasion. As noted by the authors, their results are different from previous studies that showed Plekha5 knockdown prevents blood-brain barrier transmigration and invasion; however, the functional assays investigating the role of Plekha5 presented in this work in figure 6 are convincing.

We would like to thank Reviewer 2 for the appreciation of the solidity of our work.

Major comments:

1) Bulk whole-exome sequencing of 23 breast tumors from Brca1-deficient mice in figure 1 and supplementary figure 1 identified C>A/G>T transversion mutation as the major mutation type. However, in single-cell whole-exome sequencing data from different tumor developmental stages in Brca1-deficient mice (total n=135), C>A/G>T signature appears to constitute a relatively minor fraction (Supplementary figure 3h). Instead, C>T/G>A transition mutation is the predominant change in single-cell sequencing data. It is not clear how the authors reconcile this discrepancy. Is this mutation signature difference due to sample size or tumor stages or others?

We thank the Reviewer 2 for raising this interesting question. There are several explanations about this discrepancy. First, as the Reviewer pointed out, the Supplementary Figure 1c showed C>A/G>T transversion mutation is the major type in the bulk sample; however, C>T/G>A is the predominant change in several bulk samples, including 476PT, 153PT, 153LMT, 1867-5L, etc. The single-cell data also showed C>T/G>A transition mutation is predominant change because the single tumor cells were from 476PT, 153PT, 153LMT, this observation, in fact, showed consistency between bulk and single-cell data. Second, it is very interesting that the samples 476PT, 153PT, 153LMT, 1867-5L belong to p53 wildtype group (*Brca1*^{co/co};MMTV-Cre;p53^{+/+}),

while others are p53+/- except for 1949-3L, so the p53 status might be a co-factor that affects the transversion mutation. Third, single-cell samples from different tumor developmental stages showed a similar transversion mutation pattern in Supplementary Figure 3h, so the tumor stage may not be the factor affecting the transversion mutation. To avoid causing confusions to the readers, we adjusted the main text in the revised manuscript (Page 5, line 122-125).

Overall, this is an interesting study that could be acceptable for Nat Comm. I have only a few comments.

Minor comments:

1) In the introduction section of the manuscript on page 4, the authors described that “Single-cell whole-genome sequencing has been used for copy number variation (CNV) analysis in several studies”. The authors should cite seminal work from this field. The same is true for this statement in the same paragraph: “Whole-exome sequencing was previously used to obtain high coverage for confident SNV calling in single-cell analysis”.

Some seminal works of this field have been cited in the revised manuscript. Reference 22 and 23 were cited for “Single-cell whole-genome sequencing has been used for copy number variation (CNV) analysis in several studies”, and reference 24 and 25 were cited for “Whole-exome sequencing was previously used to obtain high coverage for confident SNV calling in single-cell analysis”.

2) On page 5, it should be Wellcome Trust Sanger Institute instead of Welcome Trust Sanger Institute.

We thank the Reviewer for noticing this typo. The typo has been corrected in the revised manuscript (Page 5, line 129).

Reviewer #3 (Remarks to the Author):

The authors are using both bulk and single cell WES of BrCa1 deficient mammary glands and tumors to identify SNV and CNV that might be associated with tumor initiation. The manuscript is well written and moderately easy to follow, but is dense with data that makes reading the paper a long process. Overall, this is a detailed and comprehensive analysis of subsequent mutations that develop in BrCa1 mutant tumors that advances our knowledge of the tumor-initiating events subsequent to having a germline mutation in BrCa1.

We would like to thank Reviewer 3 for the appreciation of the solidity of our work.

Comments:

1. Introduction, page 3: In the first paragraph there is a sentence stating that drug resistance can be a problem for PARPi effectiveness. This sentence should be prefaced by a new sentence stating that PARP inhibitors have been found to be very effective in control of BRCA1 mutated tumors, however, drug resistance can still develop and reduce effectiveness of this therapy.

We thank the Reviewer for pointing this out. We have added a new sentence in the revised manuscript (Page 3 and lines 66-69):

“The poly (ADP-ribose) polymerase inhibitors (PARPi) have been approved by the U.S. Food and Drug Administration (FDA) for the treatment of *BRCA1* mutated tumors^{10, 11}; however, only a portion of patients respond to this treatment^{10, 11, 12} and drug resistance can still reduce the effectiveness of this therapy¹³.”

2. In general, many figures are so detailed and small that they are almost impossible to decipher, even when enlarged. For example, Figures 1c and d, Supp Figures 2b and 3f.

In the revised version, we reorder the Supplementary Figure 3f of the old version to Supplementary Figure 3d. We have adjusted the Supplementary Figures 2b and 3d to show images with high contrast or only several representative images in the revised manuscript. In addition, because it was limited by the size of the file in the peer-review process, we also have uploaded the original figure of Figure 1 in a separate file, which is of high quality with big size and should be able to be deciphered for Figure 1c and d when enlarged.

3. The H&E images in Figure 2c do show different morphologies but the arrows pointing to different features are not convincing. To convince the reader about the large number of neutrophils, the authors need to stain with a neutrophil specific marker. Aggressive mesenchymal tumor cells are not obvious to me in the second image. The third arrow points to an “amorphous tumor cell”. What is that? Finally, I’m not convinced by what the fourth and fifth arrows are showing. In summary, I do not find that the conclusions made from these H&E images are convincing.

The purpose of Figure 2c is to show different morphology observed in the primary tumor and liver metastatic tumor. We tried to use arrows to point out some distinct cell types but because of low magnification of the picture and without molecular markers, this did not show well. When we meant “amorphous tumor cell”, we tried to show mesenchymal type cells, but it also did not show well without molecular markers. Sorry about the confusion we caused. Because the Figure 2c is already very busy, we decided to illustrate 2 points in the bottom panel of Figure 2c. First point: mesenchymal type cells. These cells are usually spindle-shaped and Vimentin positive. We use an arrow in the first H&E image to point this type of cells, and then stain the adjacent section with Vimentin (a marker for mesenchymal cells). Actually, there are a lot of mesenchymal type of cells (the second arrow). Second point: 153 liver metastasis. The tumor cells are obviously revealed by H&E. We used arrows to point hepatocytes and also use HepPar1 to stain hepatocytes. These data have been included in Figure 2c in the revised manuscript.

4. Page 13 and Figure 2d. The authors show IHC images for the presence of proteins identified as carrying mutations. The antibodies are positive for the proteins, but are these antibodies specific for the mutant form of the protein? If not, then the detection of the protein does not “suggest that these mutations may indeed act as driving forces for the progression of these tumors”. The wild type protein is likely to be detected as well.

We thank the Reviewer for pointing this out. These antibodies are not specific for the mutant form of these proteins. What we detected are wild type proteins although expression of some of these proteins might also promote tumorigenesis. We have adjusted the text by deleting this claim in the revised manuscript.

5. Page 13: How similar was 153RT to 153PT? The statement is made that these two tumors have different driver genes but I can't find the data for 153RT. Does this tumor carry the *Plekha5* mutation or the *Arhgef11* mutation?

The morphology of 153RT (recurrent tumor) and 153PT (primary tumor) is different. This is primarily because the 153PT contains all different types of tumor cells and the 153RT might come from most aggressive cells. But because we didn't sequence 153RT, so we could not tell if 153RT contains *Plekha5* mutation or the *Arhgef11* mutation. We would like to remove the H&E of 153RT and focus on the comparison between 153PT and 153LMT as described in the first paragraph of page 14 (line 235-246). Basically, we believe 153LMT is derived from 153PT. Both tumors carry *Arhgef11* and *Plekha5* mutations revealed by single-cell DNA-sequencing, indicating their common origin.

6. Supp Figure 3c: please mark the lanes. The explanation of each lane is not clear to me.

We have marked the lanes and adjusted the Figure legend in Supplementary Figure 3c in the revised manuscript (Page 16, line 303-305).

7. Y axis of figure 4A should be labelled “Number of nSNVs”

We thank the Reviewer for noting this typo. We have changed the label of Figure 4A in the revised manuscript.

8. Following bulk sequencing, the authors have moved on the single cell sequencing, and imply in the text on page 14 that they have generated another mouse model of BrCa1 defective tumors, this time with fluorescence markers to enable BrCa1 WT and BrCa mutant cells to be detected within the same tumor. However, when labelling the resulting tumors, the numbers are the same as those used for the bulk sequencing, so I assume that even for the bulk sequencing, the authors used the mT/mG mice? This is not made clear in the text.

The Reviewer is right that several bulk-sequencing tumors are also from mT/mG mice. We did both bulk and single-cell sequencing for 3 tumors, including 476PT, 153PT, and 153LMT, which are from *Brca1^{co/co};MMTV-Cre;mG/mT* mice; but other bulk tumors are all from mice without mT/mG. We described these three bulk tumors in the bulk sequencing part. We have added the mT/mG information into the main text of bulk sequencing part to clarify this in the revised manuscript (Page 5, line 118).

9. I’m not convinced that Figure 3e is necessary.

We have deleted the figure 3e of the old version in the revised manuscript.

10. Page 24: the sentence beginning “Gbp4 was reported to be involved in the innate immune system...” needs to be re-written as it indicates that IFN γ negatively regulates IFN α , rather than Gbp4 regulating IFN α .

We have re-written this sentence in the revised manuscript (Page 24, line 430-431).

11. Page 24: the authors go on to suggest that Gbp4 may drive tumor initiation and formation, but where is the evidence that the mutation in Gbp4 has functional consequences – that the mutation increases the activity of the protein?

We have deleted this sentence to avoid this strong claim in the revised manuscript.

12. Page 24: It is surprising that a mutation Plekha5 was not detected by bulk sequencing when it was contained in 50% of the 153PT cells assessed at the single cell level. Do the authors have an explanation for this discrepancy?

We think the difference of sensitivity between bulk and single-cell sequencing is the major cause of this discrepancy. The sensitivity of bulk sequencing can be reduced by several factors. Here we feel tumor purity and heterogeneity might cause this. For tumor purity, tumor tissue is mixed of tumor cells, immune cells, and stromal cells. For tumor heterogeneity, different parts of tumors might contain tumor cells from different origins, therefore different mutation. When we isolated samples for sequencing, we only took a small piece of tumors, while saving the remaining portion for some other purpose. It is possible that the portion we took for this tumor contains very few cancer cells that contain Plekha5 mutation. For bulk sequencing, when a mutation was presented at a low level (<10%), it will be filtered out by our system. When we conduct single-cell sequencing, the portion of samples is usually bigger to include more cancer cells. The disassociation into single cells will also even out the heterogeneity issue. We also only select cancer cells for sequencing. All these factors make the data from single-cell sequencing much more reliable than bulk DNA-sequencing. Therefore, the advantage of single-cell sequencing to identify mutations was highlighted by analyzing the factors that affect the sensitivity of bulk sequencing.

13. Page 25: The authors note that Plekha5 was detected on 0.5% of BC patients in TCGA. Did this correlate with patients who also carried the BrCa1 mutation?

Because the number of *BRCA1* cancers in TCGA database is small (76 tumors), *PLEKHA5* and *BRCA1* are mutated mutually exclusive in breast cancer patients from TCGA dataset, so there is no correlation between *PLEKHA5* mutation and *BRCA1* mutation in this dataset.

14. Supplementary Figure 5j-m: What is the significance of the different colors of the bars in these graphs?

Here we want to use different colors to emphasize different points. The blue emphasizes the P value is <0.05 in the DAVID-KEGG pathway analysis, the grey indicates the P value is >0.05 , and the red emphasizes the driver genes we describe in detail in the main text.

15. What was the rationale for choosing a functional study on *Plekha5* instead of *Arhgef11* or one of the other more prominent somatically mutated genes?

Plekha5 and *Arhgef11* are two genes that more frequently mutated in single cells and can both be detected in the primary tumor and metastatic tumor. In addition, the function of these two genes is relatively less known than others. In fact, we did functional studies for these two genes, as we showed knockout of *Plekha5* promotes tumor metastasis but has no effect on tumor growth. When we knock outed *Arhgef11* in 4T1 cell, which has copy-number gain of *Arhgef11*, the tumor growth was inhibited. The data was included in Figure 5h-j and Supplementary Figure 6a and b in the revised manuscript (Page 33, line 607-614).

16. Can the authors clarify the difference between the OS curves for BC patients in Figure 6i and SuppFigure 6j. I assume that these data are from different cohorts of patients, but the outcome data for those with “normal” expression of *Plekha5* are very different for the two cohorts. In SuppFigure 6j, there are no survivors in either group after 4000 days. This is unusual for BC, unless this was a cohort with metastatic cancer at diagnosis.

We thank the Reviewer for raising this question. There are several differences between the OS curves for BC patients in Figure 6i and SuppFigure 6j in the old version: 1) As the reviewer mentioned, these data are from two different cohorts of patients, one from GEO (accession number GSE9893) and one from TCGA; 2) The expression level of *PLEKHA5* in GEO dataset is based on the microarray data, while the expression level of *PLEKHA5* in TCGA is based on NGS data, so it's different when performing OS analysis. We have mentioned this in method part of the old version, "the expression of *PLEKHA5* for OS analysis was defined as \log_2 expression value in GEO dataset, the expression of *PLEKHA5* for OS analysis was defined as z-score in TCGA dataset"; 3) For the cohort of patients from GEO dataset, as described in the original paper (Chanrion M, *et al. Clin Cancer Res.* 2008), all the patients were treated with tamoxifen (20mg daily) for 5 years after surgery, but not for the patients from TCGA data. In fact, there are survivors after 4000 days in SuppFigure 6j of the old version, we showed 10-year (3650 days) data in this figure.

Because the main purpose of our study is to demonstrate that *PLEKHA5* is a tumor metastasis suppressor. We already use a cohort of patients from GEO to support it. So, we would like to delete the TCGA dataset analysis in SuppFigure 6i and SuppFigure 6j of the old version in the revised manuscript, which will not affect our conclusion. The number of metastatic breast tumors is quite small in TCGA database, the validation of the role of genes associated breast cancer metastasis (such as *PLEKHA5*, etc.) using TCGA dataset could be done when the number of cases increases in the future.

17. Can the authors speculate on how *Plekha5* suppresses metastasis? They have shown increased migration and invasion, but is anything known about pathways in which *Plekha5* functions, especially in light of the recent paper from Hongyi Zhang et al (2020) *Cancer* 126(5):1016-1030?

We thank the Reviewer for raising this question. The recent study done by Hongyi Zhang et al showed that *PLEKHA5* positively regulates tumor growth by promoting the

G1/S cell cycle transition through its inhibition effect on *PDCD4* partially through S6K in melanoma cells. It was suggested that *MAPK/ERK* and/or *PI3K/AKT-mTOR* signaling could regulate or coregulate *PDCD4*, but It is worth noting that this regulation by one pathway versus the other seems to be context-dependent (Cuesta R et al. *Oncotarget*. 2016; Lankat-Buttgereit et al. *Biol Cell*. 2009).

We know that *PLEKHA5* is a protein containing PH domain and can bind to phosphoinositide (Simon Dowler et al. *Biochem. J*. 2000; Kenichiro Yamada, et al. *Gene*. 2012), such as PI3P, PI4P, PI5P, PI (3,5) P2 etc. Phosphoinositide plays important roles in lipid signaling, cell signaling and membrane trafficking. PI3Ks are a family of enzymes capable of phosphorylating phosphatidylinositol. We speculated that the *PI3K-AKT* pathway may be involved in the breast cancer metastasis upon loss of *PLEKHA5*. In contrast to Hongyi Zhang's study, knockout of *PLEKHA5* may activate the *PI3K-AKT* pathway in breast cancer scenario, which need to be further validated. In addition, PH domain was reported to regulate small GTPases (M.A. Lemmon. *Biochemical Society Transactions*. 2004.). So, another possibility is that small GTPase may be involved in the breast cancer metastasis process upon loss of *PLEKHA5*. All the speculations need to be further validated by experiments in the future. We have discussed this issue in the "Discussion" in the revised manuscript (Page 42, line 812).

18. There are other studies analysing the genomic changes subsequent to BrCa1 loss in breast cancer, but probably not in the detail described here. However, these previous studies should be included in the Discussion, especially the papers from J. Jonkers and colleagues.

We thank the reviewer for pointing this out. We have included some studies, including the one published by J. Jonkers and colleagues in the "Discussion" in the revised manuscript (Page 39, line 737-744) (Reference 27 and 59).

REVIEWERS' COMMENTS:

Reviewer #1 (Remarks to the Author):

I am glad that authors have addressed most of the issues raised in previous round of the review. My major concern is regarding the size and organization of figure panels. Especially, the images showing IHC and IF are too small to appreciate subtle differences. I will recommend that authors pay special attention to figure panels: Sup Figure 2B & 2C, Sup Figure 3A & 3D and, Figure 6A, B, C, D, F, G & H.

The use of term "different developmental stages" (line 248 and elsewhere) can be confused with mammary gland developmental stages. Therefore, it is recommended that tumor development or cancer progression is mentioned.

Overall, I am satisfied with the analysis and the discussion of results. Although, the characterisation of molecular mechanisms by which Plekha5 plays the role of metastasis suppressor is not addressed in this study, however, I believe that the identification of Plekha5 as a metastasis suppressor gene is an important discovery in breast cancer field.

Reviewer #3 (Remarks to the Author):

The authors have been diligent in responding to comments from all three reviewers and have modified the manuscript where appropriate.

I have just a few minor comments/questions.

1. Figure 1b: the authors show the different mutation signatures in 23 BrCa1 mutant tumors. They then state in the text that they observed different cancer signatures in this panel of 23 tumors and have text above the figure giving numbers to the different signatures. All this is presented without explanation of how the signatures were obtained and what these different numbers (0.071, 0.118 etc) mean.

2. Supp figure 4b. correct spelling (alternations is written instead of alterations).

3. Page 37: It would help the reader if cell lines B477-GFP and G600-GFP were stated to be mouse tumor lines, even though this is made clear in Methods. When I saw that they were injected into nude mice, I initially assumed that they were human breast cancer lines.

4. Page 37: line 692: The transwell assays showed that KO of PLEKHA5 promotes migration and invasion, not metastasis.

5. Figure 6, panels k and l: In response to Reviewer #1, the authors discuss a breakdown of the cohort of breast cancer patients into the different subtypes and point out that only for the Her2+ group were there sufficient numbers to show any relevance of PLEKHA5. The breakdown of their data indicates that, rather surprisingly, 111 of the total of 155 tumors were of the Her2+ subtype. This raises the question of whether PLEKHA5 is generally a metastasis suppressor of breast cancer, or only in the Her2+ subtype. I think that the authors should address this point in the text.

6. Supp figure 5j-m: The authors have explained to me the significance of the different colored bars in these figures, but should add this explanation to the figure legend as well.

REVIEWERS' COMMENTS:

Reviewer #1 (Remarks to the Author):

I am glad that authors have addressed most of the issues raised in previous round of the review. My major concern is regarding the size and organization of figure panels. Especially, the images showing IHC and IF are too small to appreciate subtle differences. I will recommend that authors pay special attention to figure panels: Sup Figure 2B & 2C, Sup Figure 3A & 3D and, Figure 6A, B, C, D, F, G &H.

We have reorganized the figures, including Sup Figure 2B & 2C, Sup Figure 3A & 3D and, Figure 6A, B, C, D, F, G &H, to make them bigger for deciphering the differences.

The use of term “different developmental stages” (line 248 and elsewhere) can be confused with mammary gland developmental stages. Therefore, it is recommended that tumor development or cancer progression is mentioned.

We have changed “different developmental stages” to “different tumor developmental stages” where it applies.

Overall, I am satisfied with the analysis and the discussion of results. Although, the characterisation of molecular mechanisms by which Plekha5 plays the role of metastasis suppressor is not addressed in this study, however, I believe that the identification of Plekha5 as a metastasis suppressor gene is an important discovery in breast cancer field.

Thank you very much for the positive comments.

Reviewer #3 (Remarks to the Author):

The authors have been diligent in responding to comments from all three reviewers and have modified the manuscript where appropriate.

I have just a few minor comments/questions.

1. Figure 1b: the authors show the different mutation signatures in 23 BrCa1 mutant tumors. They then state in the text that they observed different cancer signatures in this panel of 23 tumors and have text above the figure giving numbers to the different signatures. All this is presented without explanation of how the signatures were obtained and what these different numbers (0.071, 0.118 etc) mean.

The signatures were obtained by using a method/software, *deconstructSigs* (Rosenthal R, et al. *Genome Biology* (2016)), which allows the identification of mutational signatures within a single tumor based on 30 published mutational signatures in human cancers (Nik-Zainal S. et al., *Cell* (2012); Alexandrov L.B. et al., *Cell Reports* (2013); Alexandrov L.B. et al., *Nature* (2013);

Helleday T. et al., Nat Rev Genet (2014); Alexandrov L.B. and Stratton M.R., Curr Opin Genet Dev (2014)). We have described this in the 'Mutational signature analysis' of the 'Methods' section. To make it clearer to readers, we have mentioned 'Mutational signature was generated by deconstructSigs using all base pair changes detected in samples' in the associated Figure legend in the final version of manuscript. By using deconstructSigs, signature weights (0.071, 0.118 etc) were generated to determine the contributions of each mutational process from the published 30 signatures in our cohort of tumor samples. We have also added 'Signature weights' in the figure.

2. Supp figure 4b. correct spelling (alternations is written instead of alterations).

Thank you very much. We have corrected it and any other possible typos.

3. Page 37: It would help the reader if cell lines B477-GFP and G600-GFP were stated to be mouse tumor lines, even though this is made clear in Methods. When I saw that they were injected into nude mice, I initially assumed that they were human breast cancer lines.

We have stated that the cell lines B477 and G600 are mouse cell lines in the main text and figure legend in the final version of manuscript.

4. Page 37: line 692: The transwell assays showed that KO of PLEKHA5 promotes migration and invasion, not metastasis.

Thanks. We have replaced 'metastasis' with 'invasion'.

5. Figure 6, panels k and l: In response to Reviewer #1, the authors discuss a breakdown of the cohort of breast cancer patients into the different subtypes and point out that only for the Her2+ group were there sufficient numbers to show any relevance of PLEKHA5. The breakdown of their data indicates that, rather surprisingly, 111 of 155 tumors were of the Her2+ subtype. This raises the question of whether PLEKHA5 is generally a metastasis suppressor of breast cancer, or only in the Her2+ subtype. I think that the authors should address this point in the text.

Our experiment data showing PLEKHA5 suppressing metastasis is very solid. We have used three cell lines: B477, a mouse Brca1-WT mammary epithelium cell line; G600-GFP, a mouse Brca1-MT tumor cell line; and MDA-MB-231-GFP, a human basal-like tumor cell line. The knockout of PLEKHA5 in all these cell lines could promote metastases. Regarding the current human data about PLEKHA5 in metastasis, we could only find information about PLEKHA5 and metastatic status in 155 breast cancers. Among them, Basal group has 14 tumors (no metastasis: 13; metastasis: 1), Lum A group has 8 tumors (no metastasis: 5; metastasis: 3), Lum B group has 12 tumors (no metastasis: 3; metastasis: 9), Normal group has 10 tumors (no metastasis: 1; metastasis: 9), and Her2 group has 111 tumors (no metastasis: 85; metastasis: 26). Only Her2 group contain enough number of tumors for statistical analysis, so we only analyzed Her2 group, and the data revealed significant correction between the low levels of

PLEKHA5 and cancer metastasis. We agree that the current size of the cohort is small and contains mostly Her2 subtype. The number of metastatic breast tumors or primary breast tumors with metastases is quite small in all public databases. Thus, we are not surprised that the first available cohort happened to have more Her2 type cancer than other subtypes. In addition, PLEKHA5 is currently an under-studied gene in breast cancer. We predict that with time goes on when more data about PLEKHA5 expression on cancer metastasis immerging, the information will become better reprehensive to all subtypes. Because of this limit, we feel it is not the time yet for us to touch whether PLEKHA5 is generally a metastasis suppressor of breast cancer, or only in the Her2+ subtype, which will also certainly leave room for addressing this issue in the future study when more information becomes available.

6. Supp figure 5j-m: The authors have explained to me the significance of the different colored bars in these figures, but should add this explanation to the figure legend as well.

We have added the explanation to the figure legend.